



# On the spatial variability of the regional aerosol distribution as determined from ceilometers

Matthias Wiegner[1], Alexander Geiß[1], Ina Mattis[2], Fred Meier[3], and Thomas Ruhtz[4]

[1]Ludwig-Maximilians-Universität, Meteorologisches Institut, Theresienstraße 37, 80333 München, Germany
[2]Deutscher Wetterdienst, Meteorologisches Observatorium Hohenpeißenberg, Hohenpeißenberg, Germany
[3]Technische Universität Berlin, Chair of Climatology, Institute of Ecology, Berlin, Germany
[4]Freie Universität Berlin, Institute for Space Sciences, Berlin, Germany

**Correspondence:** Matthias Wiegner (m.wiegner@lmu.de)

**Abstract.** Measurements of the vertical distribution of aerosol particles are typically only available at selected sites leaving the question of their representativeness for urban and regional scales unanswered. As a contribution to solve this problem we have investigated ceilometer signals from two testbeds in Munich and Berlin, Germany. For each testbed measurements of 24 months from 6 ceilometers were available. This constitutes a unique data set, in particular as the same type of instruments are deployed and the same data evaluation schemes applied. Two parameters are discussed: the mixing layer height (MLH) as an indicator for the vertical distribution and the integrated backscatter as a proxy for the amount of aerosols in the mixing layer. The MLH was determined by the COBOLT algorithm, the integrated backscatter from the Klett (backward and forward) inversion scheme. It was found that the mean difference of the MLH at two sites within a testbed typically only varies by less than 50 m, slightly increasing with the distance of the corresponding sites. Almost 60 % of all intercomparisons agree within ± 100 m. MLHs are typically correlated with $R > 0.9$ in particular for the Berlin-testbed. With respect to the integrated backscatter the correlation is in the range of $0.7 < R < 0.9$. This is expected from the diversity of local aerosol sources within a given testbed. We conclude from our data that the MLH determined from a single ceilometer is applicable for a whole metropolitan area. However, the integrated backscatter of particles within the mixing layer exhibits a variability of 15–25 % suggesting that one ceilometer is not representative, especially if atmospheric processes shall be investigated.

## 1 Introduction

The spatial distribution and properties of aerosol particles are relevant for studies on the radiation budget and the hydrological cycle, for the verification of chemistry transport and dispersion models (e.g., Lemonsu et al., 2006; Emeis et al., 2011; Korhonen et al., 2014; Koffi et al., 2016; Cazorla et al., 2017; Chan et al., 2018), for flight safety (e.g., Flentje et al., 2010; Wiegner et al., 2012), and for air quality studies (e.g., Schäfer et al., 2011; Geiß et al., 2017; Mues et al., 2017). Due to the large number of sources and transformation processes the distribution of aerosols is highly variable in time and space. However, due to spatial and temporal gaps this variability is difficult to be monitored by measurements. As a consequence it must often be assumed that the aerosol distribution is "relatively homogeneous", i.e., measurements at one site are representative for a larger area, especially when aerosols are the relevant atmospheric constituent and not dominated by cloud formation or precipitation. This



assumption is e.g. applied when AERONET data of the aerosol optical depth are used to validate satellite retrievals (Filonchyk et al., 2019) or numerical models (Balzarini et al., 2015; Curci et al., 2014; Palacios-Pena et al., 2018), when ground based lidar measurements from EARLINET are used to validate Calipso data (Mona et al., 2009), or when for air quality studies the vertical distribution of aerosols measured at one site inside a city is assumed to be valid for the whole municipality (Geiß

et al., 2017). In this paper we want to contribute to the topic of the spatial variability of the vertical distribution of aerosol particles. To describe the distribution by a single parameter often the mixing layer height (MLH or $z_{\mathrm{mlh}}$) is selected – it allows to separate the free troposphere (almost aerosol free unless long range transported plumes are advected) from layers in contact with the surface with high concentrations of particles. MLH is widely used though it is neither precisely defined nor a prognostic variable of numerical models.

Measurements of the vertical distribution of aerosol particles are typically based on lidar remote sensing. Such measurements provide the particle backscatter coefficient $\beta_p$ or the extinction coefficient as a function of height. In the case of so-called ALCs (automated low power lidars and ceilometers) quantitative results are confined to the backscatter coefficient (Wiegner and Geiß, 2012). Procedures for the determination of the MLH have been developed, extensively tested and intercompared (e.g., Eresmaa et al., 2006; Haeffelin et al., 2011; Geiß et al., 2017; de Bruine et al., 2017; Poltera et al., 2017; Kotthaus and Grimmond,

2018a). Most of them are based on backscatter coefficients, attenuated backscatter or range corrected signals. Statistics of diurnal and annual cycles has been published for selected sites, e.g. Vancouver (Van der Kamp et al., 2010), Vienna (Lotteraner and Piringer, 2016), Beijing (Tang et al., 2016), London (Kotthaus and Grimmond, 2018b), Leipzig (Baars et al., 2008), Dehli (Murthy et al., 2020), or Warsaw (Wang et al., 2020). The observation periods range from 3 months (Dehli) to 10 years (Warsaw).

Our main motivation for this study was our previous work (Geiß et al., 2017) when in the framework of an air quality study the MLH was used as an indicator for the vertical range where pollutants were concentrated. In that case we had to rely on ceilometer measurements at one location inside Berlin, Germany, and could only assume that these measurements were representative for the whole city.

Investigations of the horizontal variability of the MLH are quite rare, or limited to scales larger than a metropolitan area.

Exploitation of methodologies based on the wind field, temperature profile, or trace gas concentrations are frequently used to estimate the MLH but should not be used to characterize the aerosol distribution. Lotteraner and Piringer (2016) investigated differences between Vienna and a rural site in Obersiebenbrunn, Austria, 26 km east of Vienna; ceilometer data from roughly one year were available. They found larger MLH at Vienna: for summer and winter the difference at noon was in the order of 50 m. Scarino et al. (2014) compared data from a stationary ceilometer with airborne measurements over the Los Angeles area

in May 2010. They found very good agreement (mean bias difference of 10 m and correlation coefficient of $R = 0.89$) up to 30 km away from the ceilometer site, but essentially no correlation for larger distances. Pal et al. (2012) exploited a set of four days of measurements at Paris, and compared them with measurements at Palaiseau (sub-urban) and Trainou (rural), 20 km and 105 km south of Paris, respectively. They found differences between the urban and the sub-urban site of the daytime MLH in the order of 50–100 m with larger values at the urban site. Zhu et al. (2018) compared ceilometer measurements at Jiandemen,

China, with three sites in a distance of 10–20 km, two stations 55 km off, and three stations between 120 and 270 km apart





from Jiandemen. Based on data from approximately one month they found correlation coefficients of approximately $R = 0.90$ for the MLH at the urban scale. The MLH averaged over daytime agreed within 40 m, the differences of the maximum were not explicitly determined. The correlation quickly drops with increasing distance. Ceilometer networks recently established by national weather services are primarily focusing on good spatial coverage of the corresponding country and have not yet been

used for small scale investigations.

In this paper we combine measurements of ceilometers from different institutions to tackle this problem: for two testbeds in Munich (testbed M) and Berlin (testbed B), Germany, we exploited 4 ceilometers each within distances between 2 km and 30 km plus 2 ceilometers in a larger distance. This allows us to investigate – to our knowledge for the first time using a set of identical instruments – the spatial variability of the MLH on urban and regional scales: inside a metropolitan area, as well as

differences between urban/sub-urban and rural (distances of the order of 50 km) sites. We expect that our results can also be beneficial for the validation of satellite remote sensing when large pixels are compared to ground based point measurements. Intercomparisons on the basis of quantitative physical quantities, e.g. derived from the particle backscatter coefficient, have not yet been provided.

In the following we introduce the measurement sites and the instruments, and briefly describe the algorithm to determine

the MLH. The results (section 3) include intercomparisons of the MLH, of their diurnal cycle, and the particle backscatter integrated over the mixing layer. A summary and suggestions for further studies conclude the paper.

## 2   Instruments, Sites and Methods

An overview of the locations of the ceilometers is given in Fig. 1 and Fig. 2 for testbed M and testbed B, respectively, with the key parameters summarized in Table 1. Note, that 5 ceilometers are installed on platforms, their height $z_0$ above the ground

is given in the sixth column. All ceilometers used in this study are CHM15k (Lufft, OTT HydroMet) with the only exception of M-TH (see below). The micro-chip Nd:YAG-laser emits radiation at 1064 nm with a typical pulse energy of 8 $\mu$J and a pulse repetition frequency of about 6.5 kHz. The beam divergence is less than 0.3 mrad; the field of view is 0.5 mrad. The backscattered photons are detected by an avalanche photodiode in photon counting mode. Data are stored with a resolution of 4.995 m. Standard output files comprise 1024 range bins with a reduced resolution of 14.985 m. The optical overlap is corrected

by a function provided by the manufacturer for each ceilometer to allow quantitative aerosol remote sensing typically above 240 m. The ceilometer installed at LMU (M-TH) is a special version of the CHM15k with a larger field of view (1.8 mrad) and the optical axes slightly tilted resulting in a lower limit of the measurement range (about 150 m).

The instruments are owned and operated by the German Weather Service (DWD), the Meteorological Institute of the Ludwig-Maximilians-Universität (LMU), the Technische Universität Berlin (TUB), the Freie Universität Berlin (FUB), and Lufft (OTT

HydroMet).

The distances between all ceilometers of testbed M are given in Table 2. Munich has approximately 1.5 million inhabitants. The total area of Munich is 310 km$^2$ with a maximum extension of 21 km (north–south) and 27 km (east–west). M-TH and M-HW are both in the center of Munich (Theresienstraße 37 and Helene-Weber-Allee 21, respectively): they are installed on top





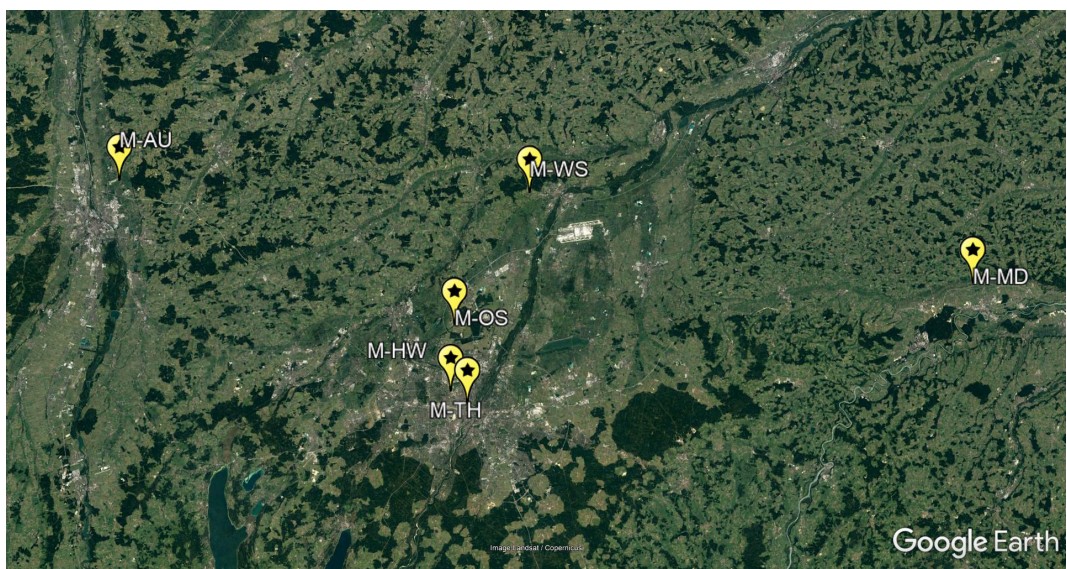

**Figure 1.** Location of the ceilometers of testbed M, see also Table 1. The distance between all sites is given in Table 2; © Google Earth Pro, 2020.

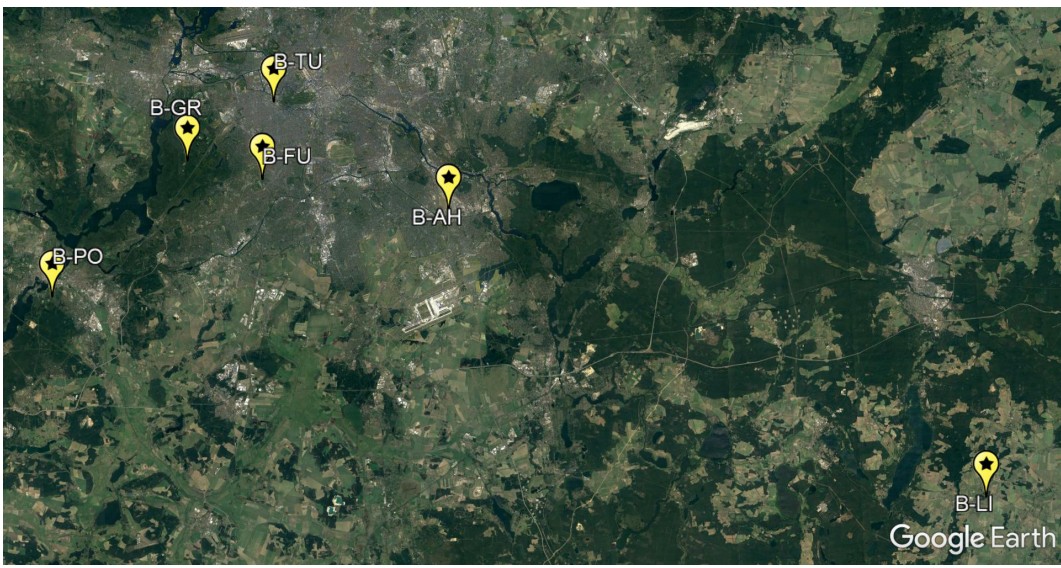

**Figure 2.** Location of the ceilometers of testbed B, see also Table 1. The distance between all sites is given in Table 3; © Google Earth Pro, 2020.



**Table 1.** List of the ceilometer sites of testbed M (upper block) and testbed B (lower block). The geographical coordinates are given in degrees. Altitudes of the ground at the ceilometer's location and the altitude of the measurement platform above the ground ($z_0$) are given in meters.

| ID | site | latitude | longitude | altitude | | owner |
|------|------|----------|-----------|--------|----------|-------|
| | | | | ground | platform | |
| M-TH | Munich, Theresienstraße | 48.1478 N | 11.5735 E | 516 | 23 | LMU |
| M-HW | Munich, Helene-Weber-Allee | 48.1633 N | 11.5433 E | 516 | 21 | DWD |
| M-OS | Oberschleißheim | 48.2443 N | 11.5525 E | 485 | 0 | DWD |
| M-WS | Weihenstephan | 48.4025 N | 11.6945 E | 477 | 0 | DWD |
| M-AU | Augsburg | 48.4253 N | 10.9420 E | 462 | 0 | DWD |
| M-MD | Mühldorf | 48.2791 N | 12.5024 E | 407 | 0 | DWD |
| B-FU | Berlin (FU) | 52.4578 N | 13.3110 E | 69 | 11 | FUB |
| B-TU | Berlin (TU Campus Charlottenburg) | 52.5123 N | 13.3279 E | 35 | 46 | TUB |
| B-GR | Berlin (TU Grunewald) | 52.4732 N | 13.2251 E | 57 | 0 | TUB |
| B-AH | Berlin (Adlershof) | 52.4302 N | 13.5247 E | 35 | 18 | Lufft |
| B-PO | Potsdam | 52.3813 N | 13.0621 E | 81 | 0 | DWD |
| B-LI | Lindenberg | 52.2094 N | 14.1284 E | 103 | 0 | DWD |

**Table 2.** Testbed M: Distance (in km) of the ceilometer sites. The IDs are according to Table 1.

| | M-TH | M-HW | M-OS | M-WS | M-AU | M-MD |
|------|------|------|------|------|------|------|
| M-TH | – | 2.8 | 10.9 | 29.7 | 56.1 | 70.6 |
| M-HW | 2.8 | – | 9.0 | 28.9 | 53.3 | 72.4 |
| M-OS | 10.9 | 9.0 | – | 20.5 | 49.5 | 70.6 |
| M-WS | 29.7 | 28.9 | 20.5 | – | 55.7 | 61.5 |
| M-AU | 56.1 | 53.3 | 49.5 | 55.7 | – | 116.7 |
| M-MD | 70.6 | 72.4 | 70.6 | 61.5 | 116.7 | – |

of a building with a height of $z_0 = 23$ m and $z_0 = 21$ m. M-OS is the radiosonde launch site of the DWD (WMO code 10838) at the northern periphery of Munich. M-WS is located in Weihenstephan north of the town Freising (49,000 inhabitants, close to Munich international airport) at a rural site. The following two sites are mainly included to investigate potential differences on a regional scale. Augsburg is the third largest city of Bavaria (295,000 inhabitants) and about 50 km west of Munich. The

5  ceilometer M-AU is set up at the local airport approximately 7 km north of the city center. In a somewhat larger distance but east of Munich the village of Mühldorf (20,000 inhabitants) is located with the ceilometer M-MD at a small airport 4 km north of the town.

The corresponding overview of the sites in Berlin (3.5 million inhabitants, 890 km$^2$, extension 38 km north-south and 45 km east–west) is given in the lower part of Table 1 and illustrated in Fig. 2. Three ceilometers are installed on platforms as indicated





**Table 3.** Testbed B: Distance (in km) of the ceilometer sites of testbed B in Berlin. The IDs are according to Table 1.

|      | B-FU | B-TU | B-GR | B-AH | B-PO | B-LI |
|------|------|------|------|------|------|------|
| B-FU | –    | 6.2  | 6.0  | 14.9 | 18.7 | 61.9 |
| B-TU | 6.2  | –    | 8.2  | 16.2 | 23.0 | 63.8 |
| B-GR | 6.0  | 8.2  | –    | 20.9 | 14.8 | 67.8 |
| B-AH | 14.9 | 16.2 | 20.9 | –    | 31.8 | 47.6 |
| B-PO | 18.7 | 23.0 | 14.8 | 31.8 | –    | 74.8 |
| B-LI | 61.9 | 63.8 | 67.8 | 47.6 | 74.8 | –    |

in the table. The ceilometers at sites B-TU and B-GR are part of the Urban Climate Observatory (UCO) operated by the Chair of Climatology at Technische Universität Berlin for long-term observations of atmospheric processes in cities (Scherer et al., 2019a) and contribute to the research program Urban Climate Under Change [UC]² (Scherer et al., 2019b). B-TU is located

in the center of the city whereas B-FU is on the edge of the center slightly outside the Ringbahn (ring of suburban trains in Berlin) on a small hill with an altitude of 20 m relative to its vicinity. B-GR is located in the Grunewald-forest and, similar to B-AH, far outside the Ringbahn. The distances between the sites are given in Table 3. B-PO is operated by the DWD and located south-west of Berlin on a small hill (Telegrafenberg, 94 m) close to the city of Potsdam (180,000 inhabitants). The ceilometer B-LI is located at the Meteorological Observatory of the DWD in Lindenberg (only 850 inhabitants), a rural site

approximately 50 km southeast of Berlin. These two stations are included to have areas "outside" Berlin, with B-PO certainly more influenced by the city than Lindenberg.

It is clear that the distribution of sites within each testbed is "as similar as possible" but not identical: Berlin is much more extended than Munich, there is not such a limited city center as in Munich, and the terrain is quite flat.

For both testbeds data from 1 September 2017 to 31 August 2019 are available (24 months). Measurements gaps are short (a

few hours or less) and rare, the only significant gaps were for the M-AU ceilometer from 11 September 2017 to 20. November 2017, for M-OS from 5 March 2019 to 20 March 2019, for M-TH from 4 June 2019 to 31 July 2019, and for B-GR from 18 March 2019 to 8 April 2019. The resolution of all data is 15 s.

The MLH is determined by means of the COBOLT-algorithm (Geiß et al., 2017). It is identical to the "meteorological" MLH ($z_{\mathrm{mlh}}$) for ceilometers on the ground, otherwise $z_0$ of the measurement platform has to be added. To avoid confusion we denote

the "MLH" from COBOLT as $z_{\mathrm{cob}}$ with $z_{\mathrm{mlh}} = z_{\mathrm{cob}} + z_0$. The retrieval is based on three functions that depend on the gradient of the range corrected signal and the temporal variability of the aerosol distribution. Their combination represents a possible solution of the temporal development of $z_{\mathrm{cob}}$ for the day given. The first function is determined from a time $t_0$ until midnight, with $t_0$ being a certain time after sunrise when a pre-scribed duration of sunshine had occurred. Furthermore, it depends on external parameters such as the vertical range, where strong signal gradients are searched, and the treatment of clouds and

precipitation. The second function is determined from $t_0$ backward in time until midnight of the previous day, whereas the third function covers the period from sunset to midnight. Subsequently, the three functions are combined with using the second for the period from 0 to $t_0$, the first from $t_0$ to sunset and the third from 1.5 hours after sunset until midnight with interpolation





between sunset and the following 1.5 hours. By varying the individual parameters of the three functions, up to 16 combined functions are calculated and the best in terms of missing jumps of the retrieved $z_{\mathrm{cob}}$ and the strength of the gradients is selected as the final solution. A lower threshold $z_{\min}$, depends on the overlap characteristics of the ceilometer. If the mixing layer is topped by clouds COBOLT is assuming $z_{\mathrm{cob}}$ as the height of the cloud base. Such retrievals are flagged (cloud-flag). In case of rain no $z_{\mathrm{cob}}$ is retrieved according to the procedure described above but $z_{\mathrm{cob}}$ is interpolated between the last MLH-retrieval before and the first retrieval after the rain event. These retrievals are also flagged (rain-flag).

As a by-product the growth rate of the (convective) mixing layer between 2 hours after sunrise and noon is determined. For this purpose the MLH is smoothed in time and the growth rate is defined as the change of the MLH between the two inflexion points. We use the growth rate to check the reliability of the COBOLT-retrieval: Cases with negative values are interpreted as indication that a very inhomogeneous and rapidly changing aerosol stratification was present that prevents a reliable retrieval. As a consequence, these days were excluded from further evaluation, such conditions are however very rare (less than 1%).

The overlap correction function can be an issue if it creates "artificial layers". They might erroneously be interpreted as $z_{\mathrm{cob}}$. Thus, a careful check for artefacts is mandatory. To avoid such artificial layers we have determined a correction of the original overlap correction function, in case of B-LI even as time dependent function. This was done by searching for meteorological situations when vertical mixing is expected to result in a homogeneous distribution of particles within a range reaching or exceeding the nominal range of complete overlap. Such conditions can be found around noon. The corresponding adaptations of the profiles are based on the slope-method as described by Hervo et al. (2016) and were accepted as correction functions provided that no artificial layers showed up when applied to different days. Using this procedure we found that the signals are trustworthy above 220 m and can be used by COBOLT. To facilitate the interpretation of $z_{\mathrm{cob}}$ at different sites we choose the $z_{\min} = 220$ m for all instruments; so also biases in the retrieved MLH are avoided.

As the ceilometer output is not synchronized in time and temporal gaps might occur a fixed grid with a temporal resolution of 2 minutes is used for all intercomparisons of the MLH retrievals.

## 3   Results

For both testbeds data from two years are available. To investigate the horizontal variability over different distances we first compare co-incident MLHs from different sites (Section 3.2) and determine diurnal cycles of the MLH (Section 3.3). Finally, a quantitative optical property of the aerosols, the integral of the particle backscatter coefficient, is discussed (Section 3.4). At first, however, we briefly outline the filtering of cloud-contaminated cases. It should be emphasized that we use the same instruments at all sites (CHM15k), the same methodology to retrieve the MLH (COBOLT), and make the same atmospheric assumptions (e.g. lidar ratio) to avoid biases in our intercomparison.

### 3.1   Cloud filtering

As we are focusing on aerosol distributions, cloudfree conditions are preferential. Under the prevailing climatic conditions in Germany this very strict requirement would however lead to a quite limited set of days that can contribute to our study. For





the description of cloudiness we use "cloud occurrence" $C$, i.e. the fraction of time when a cloud was detected in the field of view of a ceilometer, and "cloud cover" synonymously. If not otherwise stated cloud cover is determined for 24 hours a day.

The cloud cover of all clouds regardless of their altitude is denoted as $C_{\text{all}}$. In contrast, we use $C_{\text{low}}$ and $C_{\text{mid}}$ if only clouds below 2 km and 4 km, respectively, are considered. That means that for example $C_{\text{low}}$ can be zero even if cirrus clouds exist. As the criterion for the presence of a cloud we choose $cbh(1) \neq -1$ (lowermost cloud base height) with $cbh$ provided by the proprietary software of the CHM15k-instruments. For testbed M it was found that the monthly mean $C_{\text{all}}$ at different sites agrees within a few percent with very few exceptions. The annual mean is between 41 % (M-AU) and 47 % (M-MD, B-PO).

The lowest cloud cover occurs in summer (predominantly between 30 % and 40 %), whereas between December and January $C_{\text{mid}} > 60$ % leads to unfavorable conditions for the MLH-retrieval.

Consequently our selection criteria were somewhat relaxed: optically thin high altitude clouds well separated from the mixing layer, e.g. cirrus clouds, do not prohibit MLH retrievals by means of COBOLT and thus can be included in the investigation. The same is true in the case of short showers and broken cloud fields at the top of the mixing layer as long as they are not dom-

inating the aerosols. As these are "soft" criteria we have investigated different options with respect to the duration of showers, the thresholds $C_{\text{trsh}}$ of cloud covers $C_{\text{low}}$ and $C_{\text{mid}}$, and the treatment of flagged retrievals in the following sections. Only days with fog or long-lasting rain were always excluded. Such days would be useless anyway because of the strong attenuation of the ceilometer signals.

The influence of different selection criteria with respect to cloudiness may briefly be demonstrated by comparing the number

of suitable days per month for testbed M: when the quite strict condition "$C_{\text{low}} < 0.2$ at all sites and during 24 hours" has to be fulfilled for consideration, no days are found in December and January, and only 6 or 7 days per month on average for the rest of the year. This number of days is for example not sufficient to determine diurnal or annual cycles. With a relaxed criterion "$C_{\text{low}} < 0.2$ only at the selected site and during the afternoon period" the number of suitable days within two years is in the order of 20, again with lower numbers, i.e. $2-7$, for December and January. Note, that the application of the ideal condition

"no clouds at all sites for the whole day" is unrealistic as it results in only four days out of two years.

The selection criteria must not necessarily be based on the exclusion of days as describe above, it can also be based on the MLH-retrievals available with the 2-minutes resolution provided by COBOLT. By means of the cloud-flag MLH-retrievals can be identified that are not perfect in the above mentioned sense, on the other hand we had the option to investigate how the consideration or non-consideration of cloud formation at the top of the mixing layer influence the statistics of the MLH. Rain-

flagged retrievals are however always excluded, because the time and duration of rain events are likely different at different sites; moreover, an intercomparison is meaningless as the MLH is only inferred from interpolation (see above).

As "reference configuration" for the subsequent evaluation we define the following setup of selection criteria: we only consider days with $C_{\text{low}} < 0.2$ at the corresponding site to ensure – at least locally – fair weather conditions. Rain-flagged retrievals are omitted, cloud-flagged are considered. Furthermore, days with negative growth rates of the convective MLH are excluded.





**Table 4.** Testbed M: cells above the diagonal show the mean difference of MLH $\overline{\Delta_{i,j}}$ (in km) at two sites with $i$ referring to the row and $j$ to the column of the table. Cells below the diagonal show the percentage of differences within $\pm$ 100 m, $F_{100m}$. Outliers with $|\Delta_{i,j}| > 1.01$ km are removed and the reference configuration is considered.

|        | M-TH | M-HW | M-OS | M-WS | M-AU | M-MD |
|--------|------|------|------|------|------|------|
| M-TH   | –    | 0.044 | 0.076 | 0.046 | 0.073 | 0.089 |
| M-HW   | 70.3 | –    | 0.033 | 0.006 | 0.035 | 0.047 |
| M-OS   | 54.8 | 65.6 | –    | -0.028 | 0.008 | 0.017 |
| M-WS   | 50.1 | 54.7 | 57.4 | –    | 0.036 | 0.043 |
| M-AU   | 43.2 | 55.4 | 57.4 | 52.7 | –    | 0.005 |
| M-MD   | 42.5 | 50.8 | 53.6 | 52.8 | 56.8 | –    |

## 3.2 Comparison of individual MLH-retrievals

For the intercomparison of the MLH at the different sites we consider 24 hours of measurements per day. Note, that the height of the measurement platform $z_0$ (see Table 1) is added to the height derived by COBOLT. To facilitate the reading the following generic convention is introduced for two sites $i$ and $j$: $\Delta_{i,j}$ is the difference "MLH at site $i$ minus MLH at site $j$". For the sake of brevity we use for $i$ and $j$ the ID as given in Table 1 without the name of the testbed.

### 3.2.1 Testbed M

Two parameters describing the differences of the mixing layer at all sites of testbed M are summarized in Table 4: above the diagonal the mean MLH-difference $\overline{\Delta_{i,j}}$ (in km) is shown, below the diagonal the fraction of cases with $|\Delta_{i,j}| < 100$ m, henceforward referred to as $F_{100m}$, in percent. The reference configuration is considered. It is found that for all combinations $|\overline{\Delta_{i,j}}|$ is similar and small, mostly below 50 m. Exceptions occur if central Munich (M-TH) is compared to remote sites, e.g. 89 m is found for $\overline{\Delta_{TH,MD}}$. $F_{100m}$ is in general above 50% for all intercomparisons, even for the intercomparison of M-TH with the remote sites M-AU and M-MD it is only slightly lower. The maximum of 70.3 % is found if the closest sites are considered (M-TH and M-HW). It can be seen that for the three cases, when the mean MLH-difference is almost zero ($\overline{\Delta_{AU,MD}}$, $\overline{\Delta_{HW,WS}}$, and $\overline{\Delta_{OS,AU}}$) the corresponding $F_{100m}$ is not extremely large and the distance not small. In some cases larger differences partly compensate each other or the site's environment – urban or rural – dominates the evolution of the mixing layer. Due to the selection criteria the meteorological conditions can be expected to be very similar for the intercomparisons.

A detailed overview of the distribution of $\Delta_{i,j}$ is compiled in Fig. 3. In the upper right part of the figure the relative frequency distribution $n(\Delta_{i,j})$ ($i$ and $j$ for the row and column as indicated) for all combinations of sites of testbed M is shown. The distribution $n(\Delta_{i,j})$ has been normalized by the total number of cases, and the underlying resolution of $\Delta_{i,j}$ is 0.02 km with the central interval of [-0.01 km, 0.01 km[. Cases with $|\Delta_{i,j}|$ exceeding 1.01 km are not included in $n(\Delta_{i,j})$ for meteorological reasons, this concerns e.g. 0.8 % of all measurements in case of $\Delta_{TH,HW}$ and 2.4 % in case of $\Delta_{TH,MD}$. The total number of MLH-pairs $N$ is approximately $N = 113800$ for the intercomparison of M-TH vs. M-HW, and $N = 90900$ for M-TH

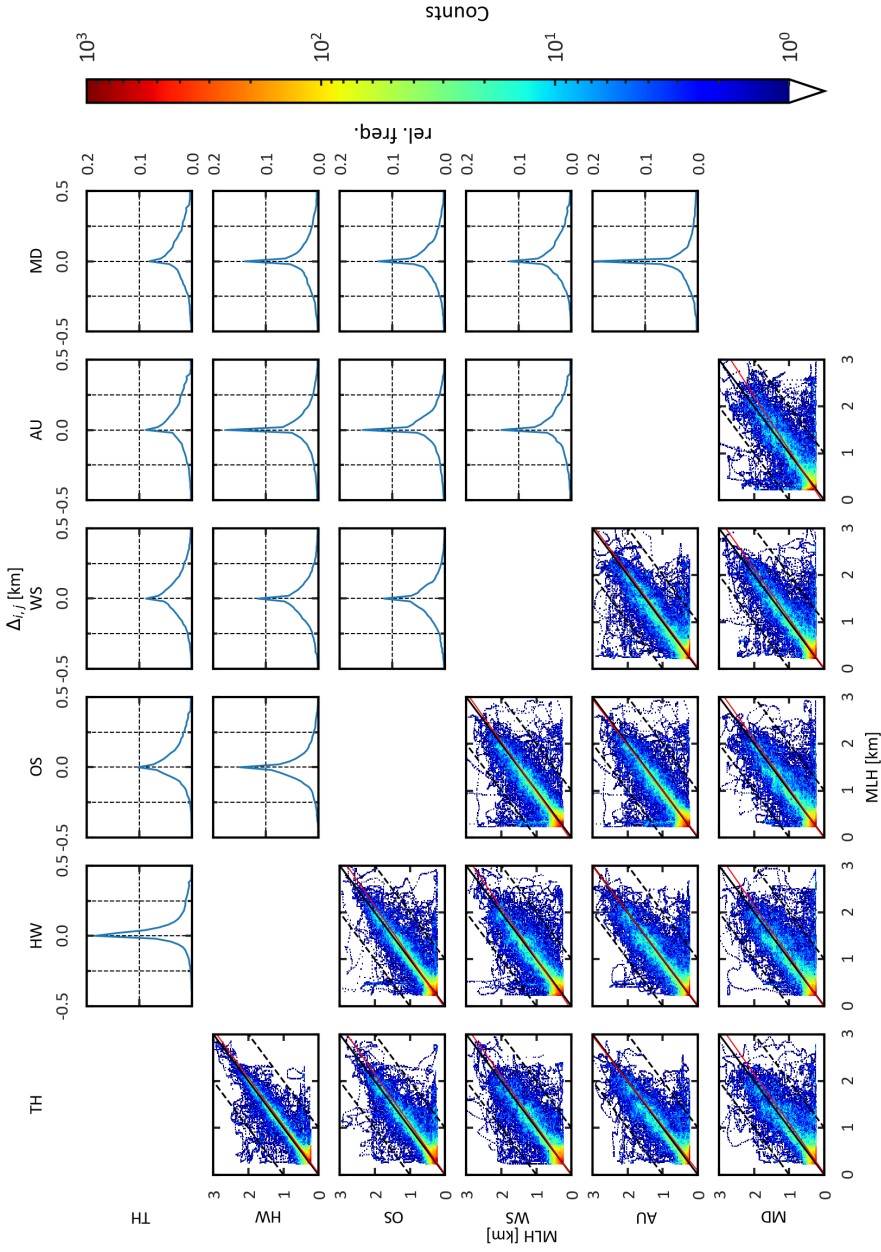

**Figure 3.** Testbed M: panels above the diagonal show the relative frequency distributions $n(\Delta_{i,j})$ for all combination of sites with $i$ as indicated by the row and $j$ by the column. The panels below the diagonal show scatter plots of co-incident MLH-retrievals (their frequency is color coded) with the horizontal axis for the site according to the row and the vertical axis according to the column. The linear regression line is plotted in red (solid line), whereas the "outlier"-regions with $\Delta_{i,j} > 1$ km are marked by the black dashed lines. The selection criteria of the reference configuration have been applied.





**Table 5.** Testbed M: Pearson's correlation coefficient ($R$, above the diagonal) and intersect $a_0$ and slope $a_1$ of the linear regression line (below the diagonal) for the reference configuration (#1). The regression line $y = a_0 + a_1 x$ is calculated with $x$ as the MLH at the site given by the column and $y$ as MLH of the site given by the row. As an example: the lower leftmost parameters are for $z_{\mathrm{mlh}}(\mathrm{MD}) = -0.051 + 0.945\, z_{\mathrm{mlh}}(\mathrm{TH})$.

|       | M-TH          | M-HW          | M-OS          | M-WS          | M-AU         | M-MD |
|-------|---------------|---------------|---------------|---------------|--------------|------|
| M-TH  | –             | 0.94          | 0.92          | 0.91          | 0.91         | 0.88 |
| M-HW  | -0.028, 0.977 | –             | 0.94          | 0.91          | 0.92         | 0.90 |
| M-OS  | -0.035, 0.939 | 0.020, 0.919  | –             | 0.92          | 0.92         | 0.90 |
| M-WS  | -0.008, 0.944 | 0.048, 0.919  | 0.061, 0.949  | –             | 0.92         | 0.91 |
| M-AU  | -0.102, 1.043 | -0.044, 1.012 | -0.023, 1.024 | -0.027, 0.986 | –            | 0.90 |
| M-MD  | -0.051, 0.945 | 0.003, 0.924  | 0.017, 0.945  | -0.002, 0.937 | 0.079, 0.863 | –    |

vs. M-MD. From the pronounced peak it is obvious that small MLH-differences are dominating (see also Table 4). As an example, $n$ of the closest sites, M-TH and M-HW, has a very pronounced maximum of 17.6 % for the $\Delta_{i,j}$-interval [-0.01 km, 0.01 km[. In the other cases with M-TH being involved $n$ becomes broader with a maximum of $8-10$ %, slightly shifted to

positive values, indicating a more extended mixing layer over central Munich (see first row of panels). Overall $n$ is "fairly symmetrical" though positive differences are dominating, when sites from the core of the testbed are related to remote sites, e.g., for the intercomparison of M-TH and M-AU or M-MD the maximum of $n$ is found for the interval [0.01 km, 0.03 km[. For all intercomparisons involving M-TH, between 64 % and 72 % of the differences are positive. Caused by the long observation period and the similarity of the locations $n(\Delta_{i,j})$ at the two most separated sites shows an unexpectedly strong peak, a very

symmetrical shape (50 % of the differences are positive), and a very small mean of $\overline{\Delta_{\mathrm{AU,MD}}} = 0.005$ km.

Below the diagonal of Fig. 3 the scatter plots of individual MLH-retrievals for each combination of sites are shown. The number of cases of a certain combination of MLHs is color coded. As already mentioned the minimum height is set to $z_{\mathrm{min}} = 220$ m. It is obvious that outliers with $|\Delta_{i,j}| > 1.01$ km, data points beyond the dashed lines, are rare. The linear regression line (red solid line) is close to the 1:1-line (black) revealing, that the same/similar MLH at the two sites can be found for both nar-

row and deep mixing layers; MLHs larger than 2 km are however not very frequent for testbed M. Most MLHs are below 500 m (red color), because nighttime measurements are included. Moreover, during winter narrow mixing layers are dominating. Further parameters derived from the scatter plots are compiled in Table 5. The values $a_0$ and $a_1$ of each regression line $z_{\mathrm{mlh,i}} = a_0 + a_1\, z_{\mathrm{mlh,j}}$ with $i$ and $j$ being the column and row, respectively, can be found below the diagonal, whereas Pearson's correlation coefficient $R$ is shown above the diagonal. In 11 out of 15 cases $a_0$ is less than $\pm 50$ m, with the largest, but still

small value for the comparison of the two remote sites M-AU and M-MD. The slope is $a_1 \approx 1$, again with the biggest deviation for the two remote sites, suggesting that extended mixing layers tend to be wider in Augsburg. The correlation coefficient does not vary much within testbed M. It is $R \geqq 0.92$ for the inner sites of the testbed – only for intercomparisons with Mühldorf it is slightly lower – but still $R = 0.88$ or larger.





**Table 6.** Settings of the selection criteria with respect to the rain-flag, filtering of unrealistic growth rates and cloud cover for 7 selected configurations

| # | rain-flag | growth rate | cloud criterion | comment |
|---|---|---|---|---|
| 1 | yes | yes | $C_{\mathrm{low}} < 0.2$ | "reference" case |
| 2 | no | yes | $C_{\mathrm{low}} < 0.2$ | |
| 3 | yes | no | $C_{\mathrm{low}} < 0.2$ | |
| 4 | yes | yes | $C_{\mathrm{low}} < 0.1$ | |
| 5 | yes | yes | $C_{\mathrm{low}} < 0.0$ | "extreme" case |
| 6 | yes | yes | $C_{\mathrm{low}} < 0.3$ | |
| 7 | yes | yes | $C_{\mathrm{mid}} < 0.2$ | |

To investigate the robustness of these findings and to estimate their uncertainty we have re-calculated the intercomparisons with different configurations, i.e. modified selection criteria. If we do not exclude MLH retrievals with the rain-flag, or if we relax the condition with respect to $C_{\mathrm{low}}$ by changing $C_{\mathrm{trsh}}$ from 0.2 to 0.3, or if we do not use the growth rate for filtering, the
number of cases for the intercomparison increases. If we choose a lower threshold for $C_{\mathrm{low}}$, e.g. $C_{\mathrm{trsh}} = 0.1$ or $C_{\mathrm{trsh}} = 0$, or if the cloud cover-criterion is changed from $C_{\mathrm{low}} < C_{\mathrm{trsh}}$ to $C_{\mathrm{mid}} < C_{\mathrm{trsh}}$ , i.e., not only cases with clouds below 2 km but also cases with clouds between 2 km and 4 km are excluded, the number of cases decreases. Explicitly we have investigated several alternative configurations as defined in Table 6: here, the consideration of the rain-flag and negative growth rates is indicated, and the cloudiness criterion.

In the case that rain-flagged MLH retrievals are not excluded (configuration #2) only about 300 additional retrievals contribute to the statistics; this was expected as rain showers are unlikely under conditions of low cloudiness (here: $C_{\mathrm{trsh}} = 0.2$). The other configurations however imply a substantial change, e.g. with the "extreme" criterion applying $C_{\mathrm{trsh}} = 0$ (configuration #5) the number of suitable cases drops from $N = 113800$ to $N = 50100$ (M-TH vs. M-HW) and from $N = 90900$ to $N = 31100$ (M-TH vs. M-MD). So the reduction of the size of the sample for the intercomparisons can be quite large.

Two examples are explicitly shown. The first example shown in Table 7 is based on the application of the "$C_{\mathrm{mid}} < 0.2$"-criterion (configuration #7) and is typical for the other configurations (except #5), so we can omit the corresponding tables here. The second example (Table 8) is based on the strict criterion "$C_{\mathrm{low}} = 0$" and constitutes the "extreme" configuration #5. From comparing the values above the diagonal of Table 4 and Table 7 it can be seen that $\overline{\Delta_{i,j}}$ typically changes by less than 10 m. The largest effect is found for $\overline{\Delta_{\mathrm{TH,MD}}}$ that decreases from 89 m to 79 m. Changes of $F_{100\mathrm{m}}$ are below 2 percent,
again the largest effect is found for the TH-MD intercomparison (from 42.5 % to 44.3 %,  see values below the diagonal). For configuration #5 without clouds below 2 km, see Table 8, the differences to the reference configuration #1 are slightly larger, especially $\overline{\Delta_{\mathrm{AU,MD}}}$ changes from 5 m to 26 m . The change of $F_{100\mathrm{m}}$ can be up to almost 5 %, for the intercomparison of M-TH and M-MD this limit is even exceeded, when $F_{100\mathrm{m}}$ increases from 42.5 % to 48.0 %. For all intercomparisons configuration #5 results in larger $F_{100\mathrm{m}}$ when compared to the reference case. The reason is that the stronger cloud filtering reduced the risk of critical MLH-retrievals.



**Table 7.** Testbed M: Same as Table 4 but with different selection criterion: $C_{\mathrm{mid}} < 0.2$ (configuration #7).

|      | M-TH | M-HW | M-OS | M-WS | M-AU | M-MD |
|------|------|------|------|------|------|------|
| M-TH | –    | 0.036 | 0.071 | 0.037 | 0.064 | 0.079 |
| M-HW | 71.6 | –    | 0.034 | 0.005 | 0.034 | 0.043 |
| M-OS | 56.3 | 65.3 | –    | -0.030 | 0.007 | 0.014 |
| M-WS | 51.2 | 54.9 | 57.2 | –    | 0.037 | 0.039 |
| M-AU | 43.9 | 55.7 | 57.2 | 53.0 | –    | 0.003 |
| M-MD | 44.3 | 51.1 | 53.5 | 54.0 | 57.1 | –    |

**Table 8.** Testbed M: Same as Table 4 but with different selection criterion: $C_{\mathrm{low}} = 0$ (configuration #5).

|      | M-TH | M-HW | M-OS | M-WS | M-AU | M-MD |
|------|------|------|------|------|------|------|
| M-TH | –    | 0.041 | 0.080 | 0.058 | 0.077 | 0.095 |
| M-HW | 73.4 | –    | 0.034 | 0.021 | 0.033 | 0.050 |
| M-OS | 55.6 | 65.7 | –    | -0.021 | 0.007 | 0.011 |
| M-WS | 54.1 | 59.6 | 59.7 | –    | 0.026 | 0.044 |
| M-AU | 46.7 | 58.7 | 60.5 | 55.1 | –    | 0.026 |
| M-MD | 48.0 | 55.1 | 57.5 | 57.3 | 57.8 | –    |

We conclude that there is no regular spatial pattern in the changes of $\overline{\Delta_{i,j}}$ and $F_{100\mathrm{m}}$ with changing configurations. From this and the fact that the definition of a configuration is somewhat arbitrary anyway, we conclude that the reference configuration (Table 4) is well suited for the investigation of the spatial variability of the MLH, and that the uncertainty of $\overline{\Delta_{i,j}}$ can assumed to be approximately 10 m, and the uncertainty of $F_{100\mathrm{m}}$ approximately 2-3 %.

It has been stated at the beginning of this section that we consider MLH-retrievals of the whole day. This includes very shallow boundary layers during night time when stable conditions prevail. As a consequence differences of the MLH at two sites are expected to be small and the correlation large. For this reason, and because convective boundary layers are of special interest, we briefly want to summarize a few key results if only the MLH from the daylight period is compared.

**Table 9.** Same as Table 4 with configuration #1, but with consideration of the daylight period only.

|      | M-TH | M-HW | M-OS | M-WS | M-AU | M-MD |
|------|------|------|------|------|------|------|
| M-TH | –    | 0.028 | 0.075 | 0.044 | 0.036 | 0.076 |
| M-HW | 70.4 | –    | 0.047 | 0.018 | 0.015 | 0.048 |
| M-OS | 53.1 | 60.6 | –    | -0.029 | -0.023 | 0.007 |
| M-WS | 46.0 | 49.8 | 53.7 | –    | 0.009 | 0.030 |
| M-AU | 42.2 | 47.3 | 50.5 | 47.6 | –    | 0.026 |
| M-MD | 39.5 | 42.4 | 47.0 | 49.8 | 46.3 | –    |



**Table 10.** Same as Table 5 with configuration #1, but with consideration of the daylight period only.

|  | M-TH | M-HW | M-OS | M-WS | M-AU | M-MD |
|---|---|---|---|---|---|---|
| M-TH | – | 0.94 | 0.91 | 0.90 | 0.89 | 0.86 |
| M-HW | 0.002, 0.965 | – | 0.93 | 0.91 | 0.91 | 0.88 |
| M-OS | -0.023, 0.940 | 0.013, 0.929 | – | 0.92 | 0.91 | 0.89 |
| M-WS | 0.002, 0.947 | 0.042, 0.929 | 0.070, 0.950 | – | 0.90 | 0.90 |
| M-AU | -0.057, 1.024 | -0.016, 1.000 | 0.021, 1.002 | 0.023, 0.962 | – | 0.89 |
| M-MD | -0.023, 0.938 | 0.010, 0.931 | 0.037, 0.946 | 0.021, 0.939 | 0.089, 0.860 | – |

10  The upper right part of Table 9 shows the mean differences $\overline{\Delta_{i,j}}$, the lower left part indicates $F_{100\mathrm{m}}$. When compared to Table 4 it can be seen that the absolute value of $\overline{\Delta_{i,j}}$ is (in 9 out of 15 cases) smaller than the 24 hours-evaluation with differences of up to 31 m in case of M-OS/M-AU. With respect to $F_{100\mathrm{m}}$ the effect of the observation period is larger: in all but one case $n(\Delta_{i,j})$ is considerably broader with a reduction of $F_{100\mathrm{m}}$ between 5 % and 10 %. The reason is that when omitting night time measurements the variability of the developing MLH before noon has a larger influence on the statistics, e.g. if the

15  onset of convection is not the same at the two sites. When comparing core and remote sites additionally the urban heat island effect may become more relevant.

Table 10 is the analog to Table 5: it shows the correlation coefficient $R$ and the parameters $a_0$ and $a_1$ of the regression line, however, for daytime measurements only. When focusing on the core sites of testbed M, the correlation coefficient is virtually unchanged, whereas for intercomparisons of core and remote sites $R$ is slightly reduced. Nevertheless, for all intercomparisons the MLHs remain highly correlated when the observation period is changed from 24 hours to daylight only. Changes of the parameters $a_0$ and $a_1$ are also very small.

### 3.2.2 Testbed B

The investigations described in the previous section for testbed M has also been made for testbed B. The total number of intercomparisons is $109400 < N < 120700$, i.e. similar to testbed M. A summary of key parameters is given in Table 11. It can be seen that the absolute value of the mean difference of the MLH is 78 m or less , in many cases (9 out of 15) even below 30 m. So the spatial variability of the MLH is lower than for testbed M. The fact that the absolute values of $\overline{\Delta_{i,j}}$ and

the heights of the buildings are of the same order of magnitude underlines the importance considering the actual altitude of the ceilometer. $F_{100\mathrm{m}}$ is in general somewhat larger than for testbed M. The values vary between 47.3 % and 79.5 %. If alternative configurations as described in Table 6 are applied the robustness of these values is confirmed.

To be consistent with the discussion for testbed M we also show all frequency distributions $n(\Delta_{i,j})$ and the MLH-scatter plots for the reference configuration in Fig. 4. The general shape of the distributions is similar. The peak of $n$ is in some cases more pronounced as can be seen in the panels above the diagonal, but not always for $|\Delta_{i,j}| < 0.01$ km. It amounts e.g. 20.7 %

for B-FU vs. B-TU with the maximum at $-0.05 \leq \Delta_{\mathrm{FU,TU}} < -0.03$ km, i.e., the mixing layer is less extended at B-FU. This might be a consequence of the topography. Small $\Delta_{i,j}$ can be found for any width of the mixing layer (panels below the





**Table 11.** Testbed B: above the diagonal the mean difference of MLH $\overline{\Delta_{i,j}}$ (in km) at two sites is shown, again with $i$ referring to the row and $j$ to the column of the table. Below the diagonal the percentage of differences within $\pm$ 100 m, $F_{100m}$, are repsented, all based on the reference configuration #1. Outliers with $|\Delta_{i,j}| > 1.01$ km are removed.

|        | B-FU  | B-TU  | B-GR   | B-AH   | B-PO  | B-LI   |
|--------|-------|-------|--------|--------|-------|--------|
| B-FU   | –     | -0.050| -0.005 | -0.019 | 0.027 | 0.015  |
| B-TU   | 70.9  | –     | 0.046  | 0.034  | 0.078 | 0.064  |
| B-GR   | 79.5  | 70.3  | –      | -0.013 | 0.029 | 0.017  |
| B-AH   | 70.4  | 65.2  | 67.1   | –      | 0.042 | 0.028  |
| B-PO   | 65.7  | 52.3  | 67.7   | 59.4   | –     | -0.013 |
| B-LI   | 55.3  | 47.5  | 54.7   | 56.8   | 54.0  | –      |

**Table 12.** Testbed B: Pearson's correlation coefficient (above the diagonal) and intersect and slope of the linear regression line (below the diagonal) for the reference configuration. The regression line $y = a_0 + a_1 x$ is calculated with $x$ as the MLH at the site given by the column and $y$ as the MLH at the site given by the row. As an example: $z_{\mathrm{mlh}}(LI) = 0.037 + 0.927\, z_{\mathrm{mlh}}(FU)$.

|        | B-FU         | B-TU          | B-GR          | B-AH          | B-PO          | B-LI  |
|--------|--------------|---------------|---------------|---------------|---------------|-------|
| B-FU   | –            | 0.97          | 0.98          | 0.97          | 0.96          | 0.94  |
| B-TU   | 0.095, 0.940 | –             | 0.97          | 0.97          | 0.96          | 0.94  |
| B-GR   | 0.026, 0.972 | -0.049, 1.003 | –             | 0.97          | 0.97          | 0.94  |
| B-AH   | 0.034, 0.979 | -0.040, 1.007 | 0.030, 0.978  | –             | 0.96          | 0.94  |
| B-PO   | 0.001, 0.963 | -0.070, 0.990 | -0.004, 0.967 | -0.007, 0.955 | –             | 0.94  |
| B-LI   | 0.037, 0.929 | -0.032, 0.959 | 0.031, 0.935  | 0.026, 0.927  | 0.068, 0.922  | –     |

diagonal), however, in contrast to testbed M, a substantial number of MLHs between 1.5 km and 2.5 km occurs. The regression line of each intercomparison (red line) is very close to the 1:1 line; its intersect $a_0$ and slope $a_1$ are compiled in Table 12. The intersect is in all cases except three (FU/TU, TU/PO and PO/LI) smaller than $a_0 = 50$ m, and $a_1$ is close to unity (numbers

5 below the diagonal). Only for intercomparisons with B-LI involved the slope is typically around 0.92–0.93, indicating that the mixing layer at Lindenberg tends to be narrower than at the other sites. The correlation coefficient is even larger than for testbed M and lies between $R = 0.94$ and $R = 0.98$. In summary the parameters $a_0$, $a_1$ and $R$ are quite similar for all 15 intercomparisons of testbed B.

The sensitivity of the intercomparisons on the selection criteria is comparable to the findings for testbed M. The influence

10 of different time periods of the intercomparison (24 hours or daylight period) is also small, for $R$ the influence is virtually negligible.

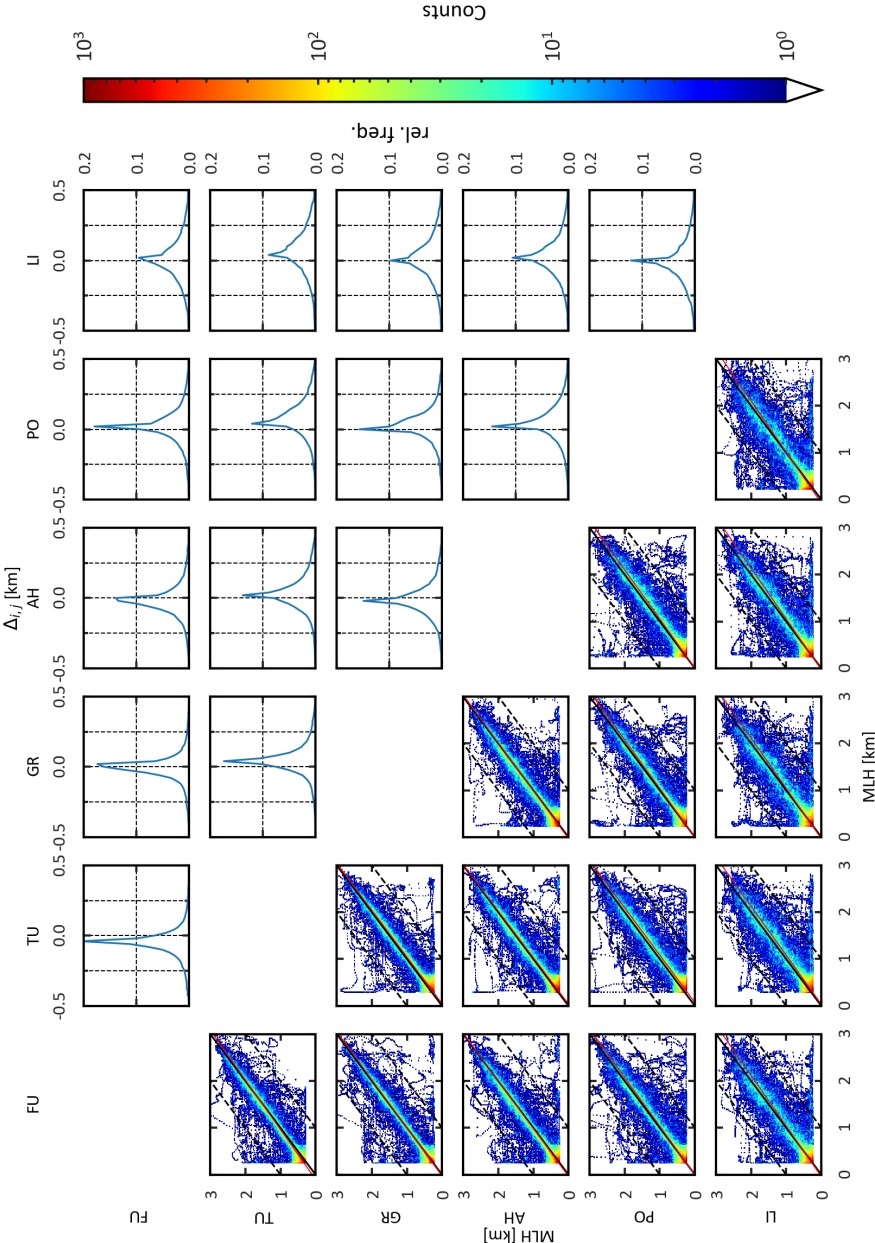

**Figure 4.** Same as Fig. 3 but for testbed B: panels above the diagonal show the relative frequency distributions $n(\Delta_{i,j})$ for all combination of sites ($i$ and $j$ according to the row/column, respectively). Panels below the diagonal show scatter plot of co-incident MLH-retrievals (horizontal/vertical axis according to the row/column, respectively) with their frequency color coded. The linear regression line is marked in red, the black dashed lines show the outlier region.





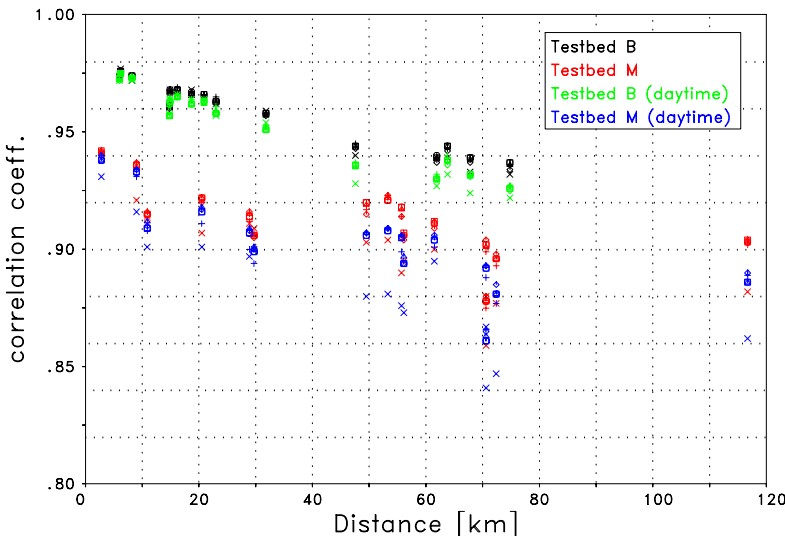

**Figure 5.** Correlation coefficient $R$ of the MLH as a function of the distance (in km) between the sites: consideration of retrievals from 24 hours for testbed B (black) and testbed M (red), and consideration of the daylight period only for testbed B (green) and testbed M (blue). Different symbols denote the different configurations (Table 6), crosses are for configuration #5 with $C_{low} < 0$, for more details see text.

### 3.2.3 Dependence on distance

To investigate the dependence of the parameters $R$ and $F_{100m}$ on the distance of the two corresponding sites we have combined the results from both testbeds. Moreover we distinguish retrievals of two different time periods: either from 24 hours or from the daylight period only. Fig. 5 concerns the correlation coefficient $R$ and shows the results from testbed M (red: 24 hours, blue: daylight period) and testbed B (black: 24 hours, green: daylight period). The different symbols refer to different configurations (Table 6) with the reference configuration marked by squares. The fact that different symbols except crosses actually cannot be distinguished demonstrate the robustness of the correlation to changes of the selection criteria. The crosses correspond to configuration #5, i.e. a very strict requirement with a substantial reduction of cases. Three main conclusions can be drawn: first, the correlation decreases with increasing distance, second, the correlation is larger for testbed B, and third, the differences between the different configurations are very small for testbed B. In case of testbed M the correlation coefficient is more sensitive to the selection of the cases, however, the uncertainty of $R$ is still small. In general the correlation coefficient is larger than $R = 0.94$ for Berlin and larger than $R = 0.90$ for Munich, when the whole day is considered. If we limit the intercomparison to measurements during the daylight period, $R$ is smaller by approximately 1 % and 2 % for Berlin and Munich, respectively. These correlations are similar to the findings close to Jiandemen, China, ($R \approx 0.90$) as reported by Zhu



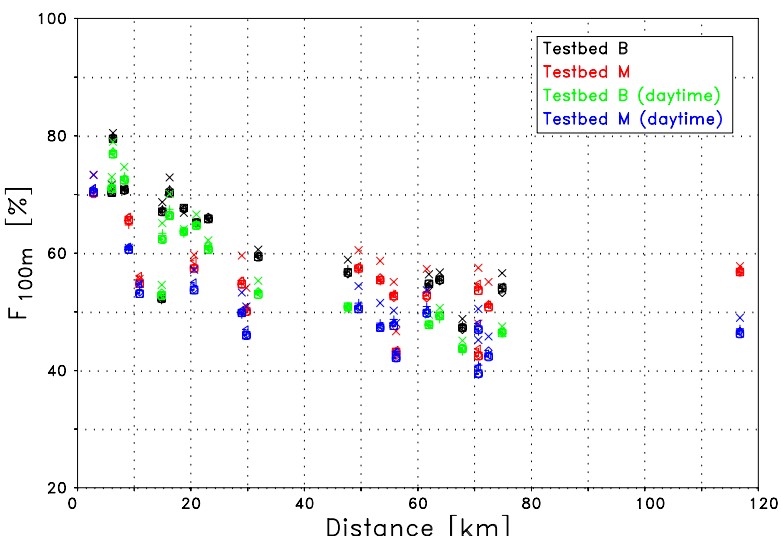

**Figure 6.** Same as Fig. 5, but fraction of cases with MLH-differences below $\pm$ 100 m ($F_{100m}$) in percent.

et al. (2018), but substantially larger than for Vienna and Obersiebenbrunn (26 km distance) with $R = 0.87$ (Lotteraner and Piringer, 2016). For distances in the order of 50 km for both German testbeds $R$ is significantly larger compared to the Chinese sites where $0.6 < R < 0.8$ was found, and the measurements in Pasadena (Scarino et al., 2014). However, only short time periods were exploited by Zhu et al. (2018) and Scarino et al. (2014), and the terrain height in California was quite variable.

The fraction of cases with MLH-differences below 100 m, $F_{100m}$, as a function of distance is presented in a similar figure
(Fig. 6). For this parameter the results for testbed M and testbed B are quite similar. The main difference is that $F_{100m}$ is slightly larger for testbed B (black and green symbols) for distances smaller than 35 km. The differences of the low values of $F_{100m}$ when only daylight observations and large distances are considered (green and blue symbols) are not considered as significant. Again, the variability is larger for testbed M, here it is in the order of the difference between the two testbeds.

### 3.3 Diurnal cycle of the MLH

For the determination of the mean diurnal cycle of the MLH we consider days selected by the criteria defined as the reference configuration, i.e. among others that $C_{low} < 0.2$ when considering the whole day. Thus, due to the relatively high cloudiness in Germany, only a limited number of days is available. As a consequence, only bi-monthly averages are determined.

The mean diurnal cycle is shown for two examples: for the April/May period it is shown in Fig. 7 as a function of time since sunrise, whereas the August/September average is shown in Fig. 8. The different colors indicate the sites, the solid lines are

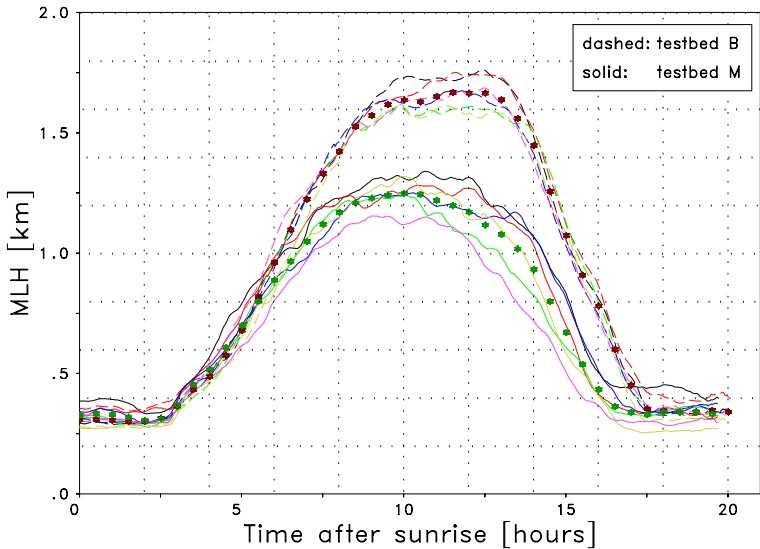

**Figure 7.** Diurnal cycle of the MLH (in km) averaged over April and May as a function of time since sunrise (in hours). The solid line is for testbed M with the different colors indicating the sites: M-TH (black), M-HW (red), M-OS (green), M-WS (blue), M-AU (yellow), and M-MD (magenta). The dashed lines are for testbed B with the sites B-FU (black), B-TU (red), B-GR (green), B-AH (blue), B-PO (yellow), and B-LI (magenta). Averages for each testbed B and testbed M are indicated by the bold dark red and green symbols, respectively.

for testbed M, the dashed lines are for testbed B. The averages for each testbed are indicated by bold red and green stars as indicated. It can be seen that the diurnal cycle at both testbeds follows the expected development, with a clear separation of all sites from different testbeds. The evolution at each site is quite similar: the onset of the growth of the mixing layer (see Fig. 7) occurs 3 hours after sunrise. When averaged from 4 to 8 hours after sunrise the averaged growth rate is 0.23 km/h for testbed B and 0.16 km/h for testbed M. The maximum of $z_{\mathrm{mlh}}$ can be found in the afternoon when the extent of the mixing

5 layer is almost constant for a few hours. For the Berlin testbed the maximum extension is significantly larger (1.67 km, bold red stars) compared to testbed M (1.25 km). The different behavior in the evening is caused by the length of the daylight period; in Berlin it is approximately 30 minutes longer than in Munich. In August/September (see Fig. 8) the difference of the maximum depth for Berlin and Munich is also large with 1.69 km and 1.34 km, respectively, and the general behavior is the same, with a slightly larger growth rate of the MLH (0.22 km/h) in the morning for testbed B compared to 0.18 km/h for testbed M. The

10 maximum bi-monthly extension of the mixing layer occurs in June/July with 1.90 km in Berlin (not shown). The spring- and autumn (October to March) averages at Munich and Berlin do not differ significantly, and in winter, December and January, the number of exploitable days is so low, that a reliable diurnal cycle cannot be determined. A clear difference between sites in the center of a testbed and remote sites is only found in case of M-MD (solid magenta line in Fig. 7) but only for April/May,

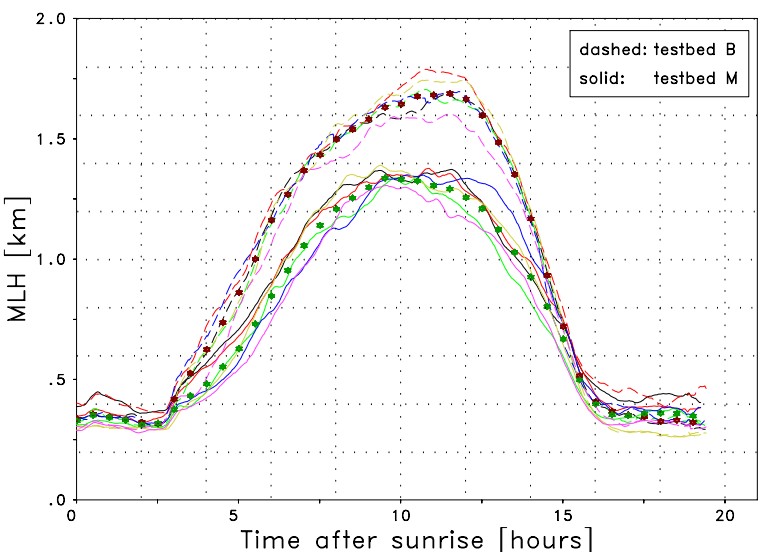

**Figure 8.** Same as Fig. 7 but averaged over August and September.

and not for August/September. For B-LI the differences to sites in central Berlin are even less pronounced so that a distinct effect of the urban heat island cannot be demonstrated.

Whether during the summer months the mixing layer in Berlin is in general more extended than in Munich should be assessed on the basis of a longer series of measurements, and a detailed analysis of meteorological parameters, topography and surface type/land use (e.g. Fallmann et al., 2016). The same is true for the effect of the urban heat island on the MLH. Only

a quick overview over relevant meteorological parameters may be provided here in Fig. 9: sunshine duration, temperature, precipitation and wind speed are shown on a monthly basis for the two testbeds. The blue curves are averaged values from two stations representing testbed M (München Stadt, München Flughafen), the orange curves are averaged over 4 stations in testbed B (Berlin-Schönefeld, Berlin-Tegel, Berlin-Tempelhof, Lindenberg). It can be seen that for testbed B in summer, when the MLH-differences are largest, the monthly mean temperature and the wind speed in 10 m altitude are larger, whereas

precipitation is lower. Moreover, Berlin experience a more continental climate than Munich. These conditions in fact point at more extended mixing layers compared to Munich. The differences of the sunshine duration are small.

The annual cycle can be estimated from the diurnal cycles, however the assessment remains critical, as there is no generally accepted definition. It can be defined as the maximum or a given, e.g. 90. or 95. percentile of the MLH during a certain period in the afternoon, e.g. between 1 or 2 hours after local noon and briefly before sunset. During this period typically the maximum

depth of the MLH in the course of the day occurs, see e.g. Fig. 7. In our case (24 months of measurements) the determination on a monthly basis is prevented by the limited number of MLH-retrievals, and a reduction of the temporal resolution is not

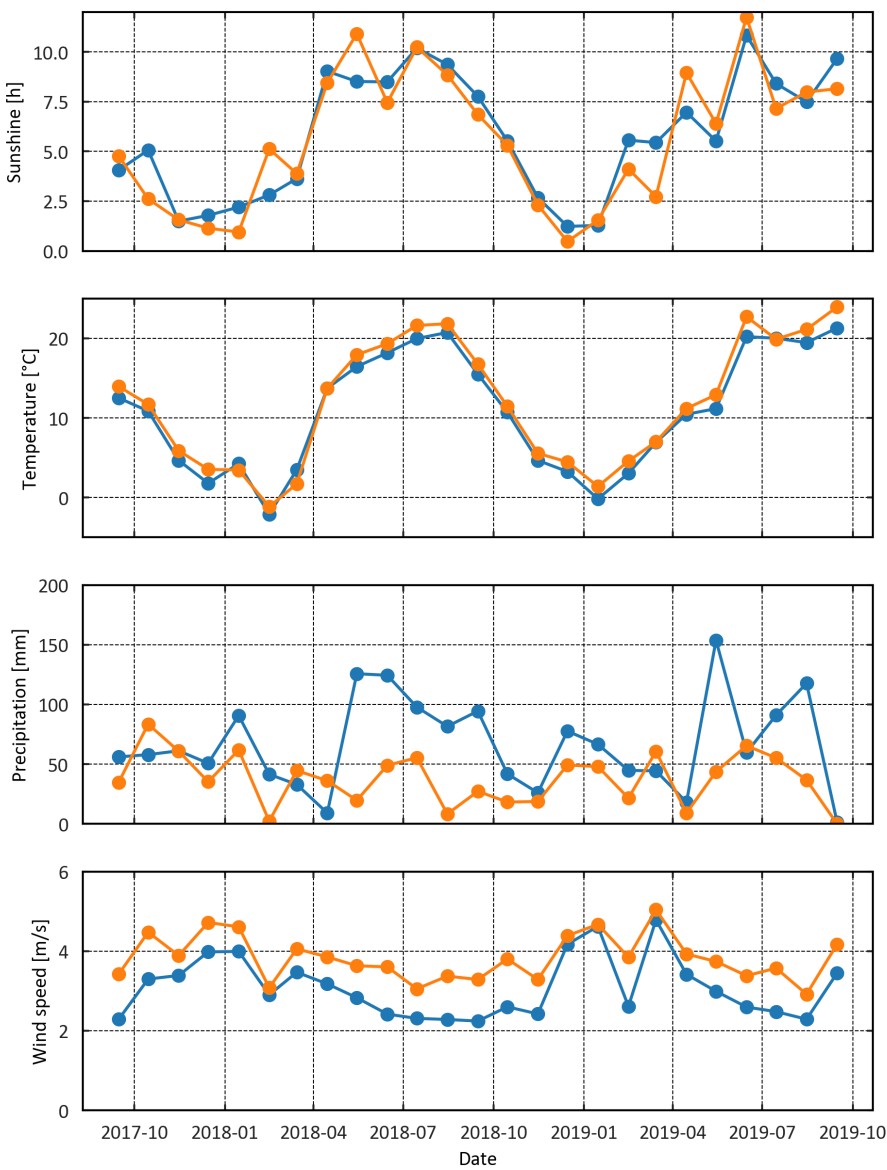

**Figure 9.** Selected meteorological parameters at both testbeds during the 24 months of observations: from top to bottom sunshine duration per month (in hours), monthly mean temperature (in degrees Celsius), precipitation per month (in mm) and monthly mean wind speed in 10 m (in m/s). Blue curves are for testbed M, red curves for testbed B. Data are taken from the Climate Data Center of the DWD (https://cdc.dwd.de/portal/)





adequate to monitor the annual evolution. Finally, it should be emphasized, that an intercomparison with other cities would suffer from the fact that publications often do not disclose how the annual cycle has been determined, or time series do not cover several years.

## 3.4 Integrated backscatter

To establish a global overview of the variability of the MLH on an urban or regional scale a general accepted definition of the MLH is mandatory. This is not yet achieved, in particular, as on the one hand there are quite different approaches possible, based on different meteorological principles, on the other hand even for retrievals based on ceilometers no standard procedure is available. As a consequence, intercomparisons of the vertical distribution of aerosols based of their optical or microphysical properties would be a important progress, in particular, as such information could facilitate the comparisons with results from numerical models. As a first step towards this direction we have intercompared the integrated backscatter $\beta_{\mathrm{int}}$ at the sites of each testbed. We define the integrated backscatter as the integral of the particle backscatter coefficient $\beta_p$ from the surface to the MLH $z_{\mathrm{mlh}}$ assuming a constant $\beta_p$ in the region of incomplete overlap, see Eq. 1.

$$\beta_{\mathrm{int}} = \int\limits_{0}^{z_{\mathrm{mlh}}} \beta_p(z)\, dz = (z_{\mathrm{min}} + z_0)\, \beta_p(z_{\mathrm{min}}) + \int\limits_{z_{\mathrm{min}}}^{z_{\mathrm{cob}}} \beta_p(z)\, dz \tag{1}$$

According to the previous investigation we set $z_{\mathrm{min}} = 220$ m, $z_0$ is given in Table 1. When $\beta_{\mathrm{int}}$ is multiplied with a mean lidar ratio one gets the aerosol optical depth (AOD) of the mixing layer at 1064 nm. In the absence of elevated aerosol layers (e.g. Saharan dust, occasionally occurring over Munich) this should be close to the AOD as determined by sun photometers. AERONET provides AOD at 1020 nm so interpolation over the small wavelength interval is not an issue in view of the uncertainty of the lidar ratio and the signal calibration.

We aim to determine the particle backscatter coefficient $\beta_p$ for all times when a MLH-intercomparison is provided. In principle, two options for the calculation of $\beta_p$ are available: we can use a backward or a forward inversion algorithm, where the former is conceptually preferential due to a better accuracy. Both approaches with a lidar ratio of 45 sr are applied in the following.

### 3.4.1 Application of the backward inversion

The temporal resolution of the MLH-retrieval, i.e. 2 minutes, cannot be kept for $\beta_p$-retrievals when the backward Klett-algorithm (Klett, 1981; Fernald, 1984) is used because the Rayleigh-calibration requires much longer time averaging. Therefore, we reduce the temporal resolution to one hour by defining 47 overlapping time slices that are centered around each half hour, i.e., the first covers the period from 00:00 to 1:00 UTC, the second the period from 00:30 to 01:30 UTC and so on. MLH-retrievals from COBOLT and ceilometer signals were averaged accordingly with cloud contaminated signals being excluded. If less than 25 % of the ceilometer profiles remain, no average is calculated. Furthermore, all cases with $z_{\mathrm{cob}} < 0.4$ km are excluded from the following investigation to avoid a strong contribution of the assumed $\beta_p$-profile in the overlap region to

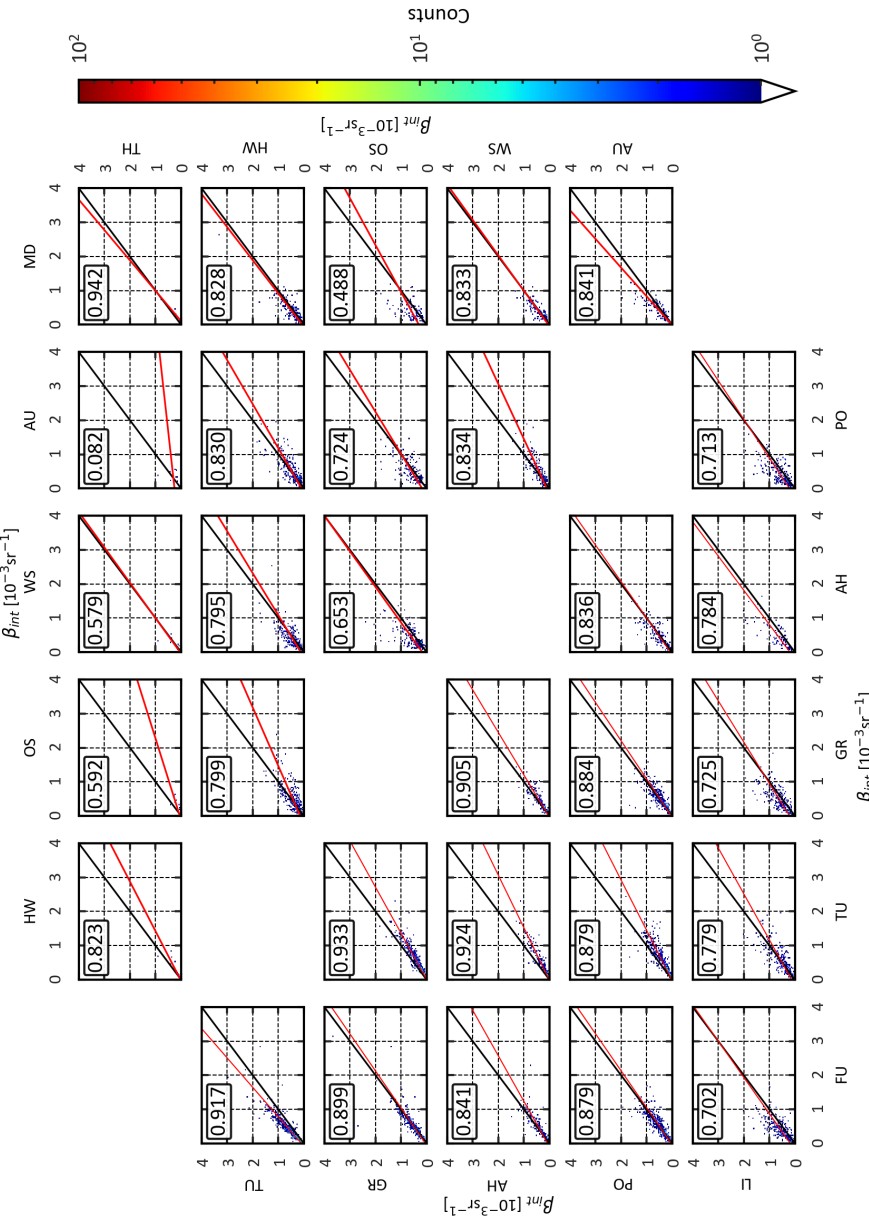

**Figure 10.** Comparison of integrated backscatter $\beta_\mathrm{int}$ for testbed M (above the diagonal) and testbed B (lower left part) as derived from the backward Klett inversion. Cases with $\beta_\mathrm{int} > 8 \cdot 10^{-3}$ sr$^{-1}$ at either site and $|\Delta_{i,j}| > 1$ km have been removed. Note, that the panels of the first row (associated to M-TH) are based on a not sufficient number of cases and are only included for the sake of completeness.





$\beta_{\mathrm{int}}$, see first term on the right hand side of Eq. 1. To have an additional criterion to prevent cloud contamination, cases with $\beta_p > 0.01$ km$^{-1}$ sr$^{-1}$ at heights $z < z_{\mathrm{mlh}}$ are excluded. As already mentioned, an refined overlap-correction is applied to the ceilometer signals.

As a consequence, the number of intercomparisons of $\beta_{\mathrm{int}}$ is much smaller than the number of MLH-intercomparisons. This is in particular true when M-TH is involved because for this instrument the Rayleigh calibration was not possible in most

cases. The reason was the aging of the avalanche photodiode occurring until June 2019 resulting in a distortion of the signal. From dark measurements, i.e. covered telescope, in December 2018 and January 2019 it was found that this distortion was stable in time, however, earlier dark measurements are not available. Correction of the ceilometer signals by means of the dark measurements increase the number of successful Klett inversions, however, the total number remains too low to make an intercomparison reasonable. For intercomparisons not including M-TH the total number is still relatively low and ranges from

124 to 327 (testbed M), and from 106 to 667 (testbed B).

The intercomparisons for the two testbeds are shown in Fig. 10. The panels above and below the diagonal refer to testbed M and testbed B, respectively. Cases with $\beta_{\mathrm{int}} > 8 \cdot 10^{-3}$ sr$^{-1}$ at either site or $|\Delta_{i,j}| > 1$ km are considered as unrealistic and thus have been removed. As mentioned above, the panels of the first row, associated with M-TH, should be ignored; they are only included for the sake of completeness. For testbed M a correlation coefficient of $0.488 < R < 0.841$ can be found; note

that the low $R$ for M-OS/M-MD is based on 124 intercomparisons only. For testbed B it is in general larger with $0.702 < R < 0.933$. In spite of the moderate to high correlation the mean difference $D_{i,j}$ (in percent) of the integrated backscatter

$$D_{i,j} = 100 \, \frac{\sum_{k=1}^{N} |\beta_{\mathrm{int},i,k} - \beta_{\mathrm{int},j,k}|}{0.5 \left[ \sum_{k=1}^{N} \beta_{\mathrm{int},i,k} + \sum_{k=1}^{N} \beta_{\mathrm{int},j,k} \right]} = 200 \, \frac{\overline{\beta_{\mathrm{int},i} - \beta_{\mathrm{int},j}}}{\overline{\beta_{\mathrm{int},i}} + \overline{\beta_{\mathrm{int},j}}} \tag{2}$$

at the two correlated sites $i$ and $j$ (with $k$ being the counter of the $N$ cases) is quite large: for testbed M $D_{i,j}$ is between 27 and 53 % (median: 32 %). Note, that in the denominator the mean of both sites is used, because the integrated backscatter is not

the same. Within the testbed the averages $\overline{\beta_{\mathrm{int}}}$ differ between $3.9 \cdot 10^{-4}$ sr$^{-1}$ at M-WS and $5.4 \cdot 10^{-4}$ sr$^{-1}$ at M-OS, revealing that the amount of aerosols (expressed in terms of backscattering) is much more variable than the MLH. For testbed B we find $14 < D_{i,j} < 44\%$ (median: 33 %), and $4.1 \cdot 10^{-4}$ sr$^{-1} < \overline{\beta_{\mathrm{int}}} < 5.9 \cdot 10^{-4}$ sr$^{-1}$. Again, the variability within the testbed is quite large. The main reason is the variability of the local aerosol sources with respect to the emission strength and the particle type. This has been shown by e.g. von Schneidemesser et al. (2018) who reported large differences of emissions within Berlin.

These differences certainly influence the aerosol content of the mixing layer, though due to the time required for then vertical mixing an immediate effect of the ground based sources on the vertical column is not expected. If we strengthen the above mentioned criteria with respect to $\beta_{\mathrm{int}}$ or $|\Delta_{i,j}|$ the number of cases is reduced but the correlation coefficient increases with the second criterion being much more effective. As an example, the correlation increases to $0.781 < R < 0.943$ for testbed B, if data with $\beta_{\mathrm{int}} > 4 \cdot 10^{-3}$ sr$^{-1}$ at either site or $|\Delta_{i,j}| > 0.2$ km are excluded.

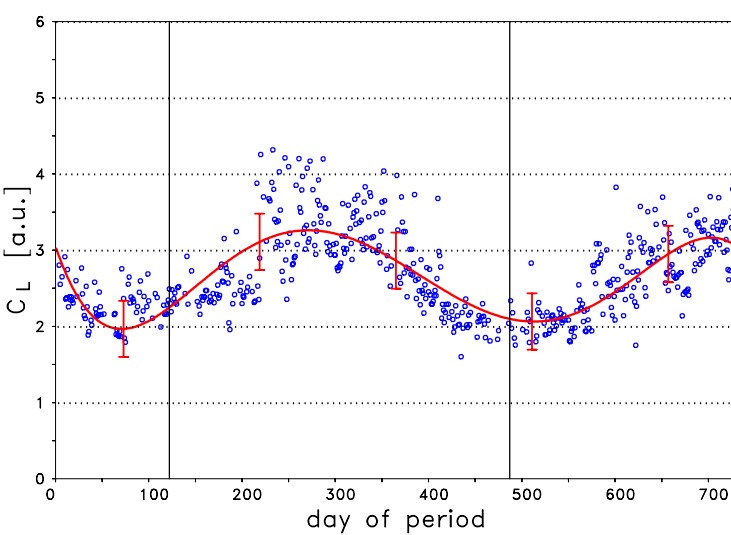

**Figure 11.** Lidar constant $C_L$ [in arbitrary units] derived from the backward Klett inversion for B-TU for the period from 1 September 2017 till 31 August 2019 on a daily basis. The vertical lines indicate the beginning of year 2018 and 2019. The red line is the 5. order polynomial fit and the vertical bars indicate the standard deviation, for B-TU 14.1 % of the mean $C_L$.

### 3.4.2 Application of the forward inversion

The main drawback of the backward inversion is the limited number of cases. Thus, we also use an alternative approach to calculate $\beta_{\mathrm{int}}$ by applying the "forward Klett inversion", i.e. instead of the determination of a boundary value of $\beta_p$ at a reference height (Rayleigh calibration) one relies on the lidar constant $C_L$ (aka calibration constant). However, it has been found that $C_L$ of most of the commercial ceilometers is not constant, consequently adding another error source to the retrieval. In principle, the lidar constant as a function of time can be determined from fitting $C_L$ from the available Rayleigh-calibrations. The advantage of the forward integration is, that $\beta_p$ and consequently $\beta_{\mathrm{int}}$ can be calculated for any time when the MLH can be retrieved, even when mid-level clouds are present or the signal-to-noise-ratio of the ceilometer signal is too low to allow a backward inversion. The disadvantage is that the accuracy of the fitted $C_L$ is limited, furthermore, the forward inversion includes the assumption that the atmospheric transmission below $z_{\mathrm{ovl}}$ is close to 1. For the wavelength of 1064 nm and low overlap ranges this assumption is justified (Wiegner and Gasteiger, 2015).

An example can be found in Fig. 11 for B-TU. It shows the daily mean lidar constant $C_L$ as averaged over all available Rayleigh calibrations of the corresponding day, taking different settings into account including 1- or 2-hour averages, different cloud filters and different width of the calibration range. All $C_L$ of the 2 year-period are then fitted by a 5. order polynomial



(red curve), the standard deviation is 14.1 % if related to $C_L$ averaged over 24 months. It can be clearly seen that the lidar constant is time dependent, with a maximum in summer and a minimum in winter. This general behavior was found for all ceilometers except M-OS that shows a more irregular temporal development (decrease) in 2019. On average the standard deviation is 15.0 %, with a slightly larger value of 20.1 % for M-OS.

Having the lidar constant as a function of time it is straight forward to calculate $\beta_p$ (e.g., Wiegner and Geiß, 2012) and

derive $\beta_{\mathrm{int}}$ for all available intercomparisons of MLHs. The forward inversion allows an improved temporal resolution of 15 minutes. As before cloud contaminated cases with $\beta_p > 0.01$ km$^{-1}$ sr$^{-1}$ at heights $z < z_{\mathrm{mlh}}$ are excluded, moreover we omit cases with $\beta_{\mathrm{int}} > 8 \cdot 10^{-3}$ sr$^{-1}$ at any of the compared sites, and cases with $|\Delta_{i,j}| > 1.0$ km. The resulting scatter plots of $\beta_{\mathrm{int}}$ and the corresponding correlation coefficients are shown in Fig. 12. It is immediately visible that the number of data points has drastically increased; the total number is between 4500 (for M-AU/M-MD) and 7500 (for B-TU/B-GR). Again the

correlation of $\beta_{\mathrm{int}}$ is larger for testbed B: the correlation coefficient exceeds $R = 0.77$ for all cases not including the rural site at Lindenberg. For testbed M, $R$ is between $0.71 < R < 0.89$. A comparison with the backward inversion is in principle difficult due to the significantly larger number of cases and the changed temporal resolution. In general $R$ is somewhat smaller, but the regression line is closer to the 1:1 line.

$\beta_{\mathrm{int}}$ is typically less correlated than the MLH (see Table 5 and Table 12 for comparison). This is caused by the uncertainty of

the calibration, in particular when applying the forward Klett inversion, and the fact that the intercomparison of the integrated backscatter is not only affected by differences of the MLH but also by differences of $\beta_p$ at the two sites. As already been demonstrated by means of the AOD over Berlin (Li et al., 2018), differences of $\beta_p$ are likely because of the spatiotemporal variability of aerosol sources and dispersion pattern triggered by the changing wind field. Therefore, the results might be influenced by a number of extreme values. Their impact has been briefly estimated by introducing various criteria to eliminate

unrealistic intercomparisons: one criterion concerns the maximum acceptable difference of the MLH at the two sites, another the maximum difference of $\beta_{\mathrm{int}}$. If we gradually decrease the first criterion from $|\Delta_{i,j}| < 1.0$ km (see Fig. 12) to $|\Delta_{i,j}| <$ 0.2 km, the number of cases decreases drastically by up to 2000, and $R$ increases. This is especially true for testbed M, where $R$ increases by $0.05-0.1$ for the different pairs of sites: the mean correlation coefficient of all 15 intercomparisons that changes from $R = 0.779$ to $R = 0.857$, may serve as an indicator for this increase. A similar behavior is found for testbed B: $R$ is in

general somewhat larger than for testbed M but the change of the mean correlation coefficient from $R = 0.809$ to $R = 0.858$ is less pronounced. The influence of the second criterion on the correlation coefficient, e.g., if we change the threshold of $\beta_{\mathrm{int}}$ from $8 \cdot 10^{-3}$ sr$^{-1}$ to $4 \cdot 10^{-3}$ sr$^{-1}$, is quite small.

The relative difference of the integrated backscatter $D_{i,j}$, Eq. (2), is between 26 % and 40 % (median: 34 %) for testbed M, and 20 % and 53 % (median: 36 %) for testbed B, i.e., quite similar to the retrieval based on the backward inversion. The

relative deviation in percent from the regression line $E_{i,j}$ for sites $i$ and $j$

$$E_{i,j} = 100 \; \frac{\sum_{k=1}^{N} |(a_0 + a_1 \; \beta_{\mathrm{int,i,k}}) - \beta_{\mathrm{int,j,k}}|}{\sum_{k=1}^{N} (a_0 + a_1 \; \beta_{\mathrm{int,i,k}})} \tag{3}$$

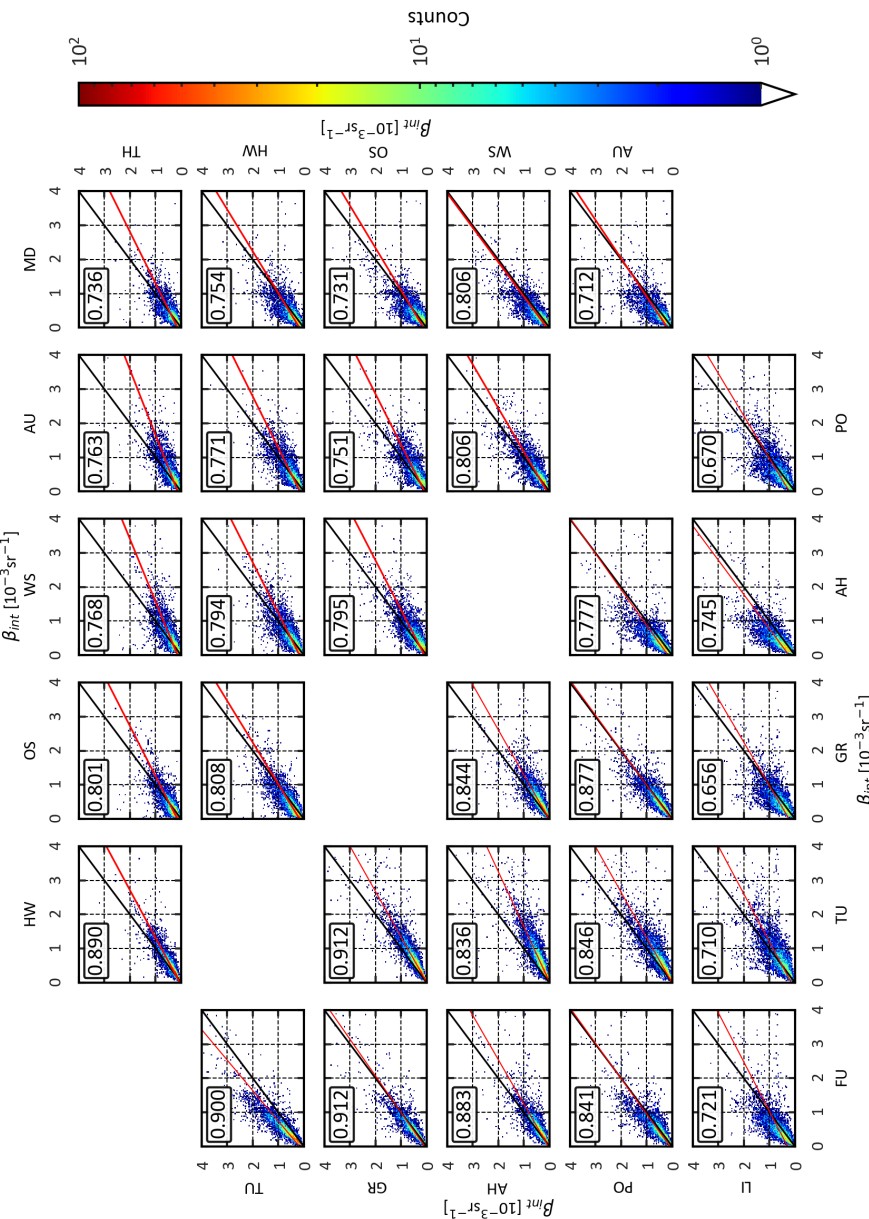

**Figure 12.** Same as Fig. 10 but $\beta_{\text{int}}$ derived from forward Klett inversions with a temporal resolution of 15 minutes. Cases with $\beta_{\text{int}} > 8 \cdot 10^{-3} \text{ sr}^{-1}$ at either site and $|\Delta_{i,j}| > 1.0$ km are excluded.





**Table 13.** Overview over $\beta_{\mathrm{int}}$ at the sites of testbed B and number of settings. The first 5 line corresponds to the application of the forward inversion, the last line to the backward inversion. The first column describes the settings: the first number is the factor applied to $C_L$ as calculated from the fitting curve, the second number is the threshold of the $|\Delta_{i,j}|$ criterion.

| settings | $\beta_{\mathrm{int}}$ [$10^{-4}$ sr$^{-1}$] | | | | | | number of cases | | | | | |
|---|---|---|---|---|---|---|---|---|---|---|---|---|
| | B-FU | B-TU | B-GR | B-AH | B-PO | B-LI | B-FU | B-TU | B-GR | B-AH | B-PO | B-LI |
| 0.85, 1.0 | 6.23 | 8.81 | 6.55 | 5.51 | 7.11 | 8.36 | 31408 | 35179 | 32519 | 32687 | 30432 | 29479 |
| 1.00, 1.0 | 5.20 | 7.37 | 5.47 | 4.55 | 5.92 | 6.97 | 31669 | 35426 | 32795 | 32919 | 30646 | 29695 |
| 1.15, 1.0 | 4.44 | 6.31 | 4.66 | 3.85 | 5.04 | 5.95 | 31908 | 35672 | 33018 | 33136 | 30850 | 29918 |
| 1.00, 0.5 | 5.18 | 7.32 | 5.44 | 4.55 | 5.91 | 6.98 | 30317 | 34039 | 31520 | 31405 | 28926 | 27479 |
| 1.00, 0.2 | 5.08 | 7.14 | 5.38 | 4.49 | 5.80 | 6.90 | 26275 | 29221 | 27163 | 26286 | 23249 | 20336 |
| bsc | 4.45 | 5.67 | 4.38 | 4.10 | 4.47 | 5.93 | 2287 | 1999 | 2212 | 915 | 1980 | 1085 |

(with $k$ being the counter of $N$ cases) is 19 % $< E_{i,j} <$ 38 % (average: 32 %, median: 33 %) for testbed M, and slightly smaller values for testbed B with 20 % $< E_{i,j} <$ 35 % (average: 28 %, median: 28 %). . The smallest deviation of 19.1 % is found for M-TH/M-HW, i.e. the closest sites of all.

To demonstrate the variability within a testbed we have also calculated averages $\overline{\beta_{\mathrm{int}}}$; here we consider all measurements used in any of the intercomparisons. The average for the sites of testbed B is 4.5 $\cdot 10^{-4}$ sr$^{-1}$ $< \overline{\beta_{\mathrm{int}}} <$ 7.4 $\cdot 10^{-4}$ sr$^{-1}$ (see

Table 13, second line). These values are higher than the previous results derived for the 1-hour resolution (last line), in particular, large differences occur if individual sites are compared. The main reason is that, due to the better temporal coverage and resolution, the samples (backward/forward inversion) of the intercomparisons are quite different. As can be seen in the right part of Table 13, the number of cases is more than one order of magnitude lower in the "backward"-case. On the other hand, the uncertainty of $C_L$ of 15 % results in an uncertainty of $\beta_{\mathrm{int}}$ of approximately 18 %: As an example the results in the

first line (underestimate of $C_L$ by 15 %) and the third line (overestimate by 15 %) may be compared to the second line where $C_L$ from the fitting curve is used. This calibration error is larger than that for the backward inversion that can be as low as 4 % (Wiegner and Geiß, 2012) whereas the additional uncertainty due to the unknown lidar ratio affects both approaches in the same way. Incidentally, an increase of $C_L$ is associated with a decrease of $\beta_p$, thus the number of cases slightly increases as potentially less cases are excluded due to the $\beta_{\mathrm{int}}$-threshold. The table also shows that the influence of different $|\Delta_{i,j}|$-criteria

is in the order of 1–3 % only.

$\overline{\beta_{\mathrm{int}}}$ of the sites within testbed M differs between 4.2 $\cdot 10^{-4}$ sr$^{-1}$ at M-TH and 6.0 $\cdot 10^{-4}$ sr$^{-1}$ at M-AU. As the conclusions do not differ from those of testbed B, a corresponding table can be omitted here.

When $\beta_{\mathrm{int}}$ is averaged over the whole testbed, we find 5.9 $\cdot 10^{-4}$ sr$^{-1}$ for testbed B and 5.0 $\cdot 10^{-4}$ sr$^{-1}$ for testbed M, corresponding to a 19 % larger value in Berlin. This is expected taking into account the more extended mixing layer in summer.

With a lidar ratio in the range of 38 to 53 sr, calculated with the online tool MOPSMAP (Gasteiger and Wiegner, 2018) for continental and urban aerosol types this corresponds to a relatively low, but not implausible annual mean AOD of the mixing layer in the range of 0.019 $< \tau_p <$ 0.031 at 1064 nm.





Finally, we include scatter plots (Fig. 13) and the correlation coefficients of the intercomparison of the mean particle backscatter coefficient $\overline{\beta_{p,\mathrm{mlh}}}$ averaged over the vertical extent of the mixing layer at the corresponding site. It can be seen that for testbed B the correlation coefficient is $0.725 < R < 0.945$, i.e. an increase compared to $\beta_{\mathrm{int}}$ (see Fig. 12). The low values are found for correlations with the remote site at Lindenberg. On the contrary the correlation of $\overline{\beta_{p,\mathrm{mlh}}}$ decreases for testbed M. This suggests that the height of the mixing layer has a stronger influence on the differences of $\beta_{\mathrm{int}}$ in Berlin than in

Munich.

In case of $\overline{\beta_{p,\mathrm{mlh}}}$ we have also determined the relative deviation $E_{i,j}$ for sites $i$ and $j$, analogously to Eq. (3). For testbed M we find $13\ \% < E_{i,j} < 30\ \%$ (average: 25 %, median: 26 %), again with the smallest deviation of 13.1 % for the closest sites M-TH/M-HW. This means that for $\overline{\beta_{p,\mathrm{mlh}}}$ the differences from the regression line are smaller than for $\beta_{\mathrm{int}}$, though the correlation is slightly lower. In case of testbed B we find $14\ \% < E_{i,j} < 31\ \%$ (average: 22 %, median: 21 %). Here, the

differences are also clearly smaller compared to $\beta_{\mathrm{int}}$. For a deeper analysis of these effects the role of e.g. the topography, the dispersion of particles from local sources and the flow pattern should be considered.

## 4   Summary and conclusions

We have investigated the spatial distribution of aerosols in two testbeds – Munich and Berlin, Germany – both of which includ-ing 6 ceilometers located inside a large city and at rural sites approximately 50 km outside the center of the city. The aerosol

distribution is described by means of two quantities that can be inferred from active remote sensing: the mixing layer height as a qualitative measure and the (particle) integrated backscatter as a quantitative measure. The first parameter can be considered as a simple description of the vertical distribution, the second as a proxy for the aerosol optical depth. Data from 24 months are available. They constitute a unique data set: the first time continuous range resolved aerosol information at several points in a regional domain could be exploited. As all instruments are of the same type, and the same methodology to determine the

MLH is applied unbiased results are provided. With respect to the integrated backscatter we use the backward and the forward inversion scheme: though the former provides a better accuracy of the signal calibration we feel that the latter is preferential as it provides a much better temporal coverage and resolution. In either case a careful overlap correction is mandatory, and consideration of the height of the measurement platform is required. With respect to the exploitation of ceilometer data efforts should focusing on improvements of automated procedures for calibration and filtering of cloud-contaminated cases should be

encouraged.

It was found that the MLH within each testbed is quite similar: typically the mean difference between two sites is below 50 m and in 50% – 60% of the cases the difference of the MLH at two sites is within $\pm100$ m. The correlation coefficient and the fraction of cases within $\pm100$ m slightly decrease with distance between the corresponding sites. However, a significantly higher MLH over the city compared to sites about 50 km outside is not found. The integrated backscatter is found to be highly

and moderately correlated within the Berlin and Munich testbed, respectively, however with mean differences of approximately 30 % when individual sites are compared. The spatial variability of the mean $\overline{\beta_{\mathrm{int}}}$ within each testbed is also considerable and amounts 15–25%. Conclusions on the horizontal homogeneity of $\beta_{\mathrm{int}}$ and $\overline{\beta_{p,\mathrm{mlh}}}$ should be based on relative values because



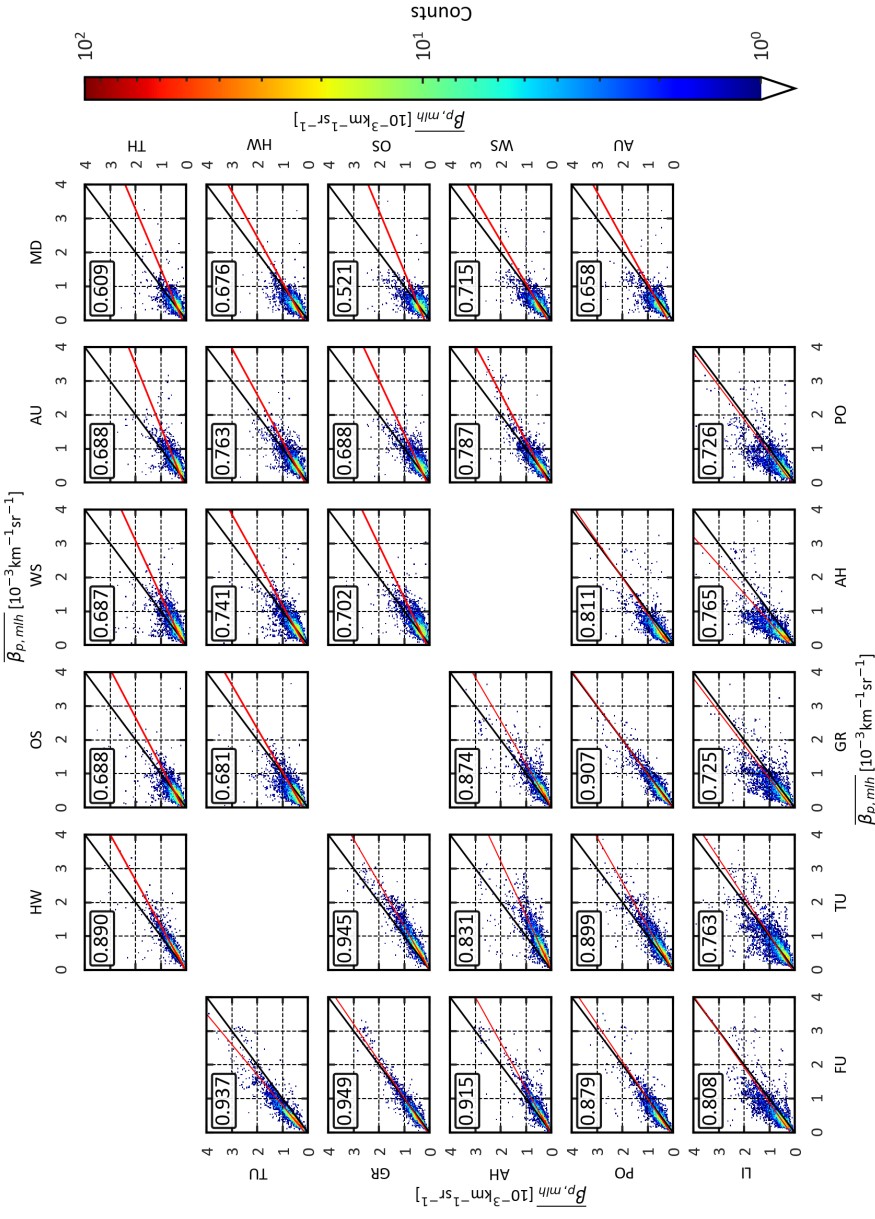

**Figure 13.** Intercomparison of $\overline{\beta_{p,\mathrm{mlh}}}$ averaged over the mixing layer as derived from forward Klett inversions with a temporal resolution of 15 minutes. Cases with $\beta_{\mathrm{int}} > 8 \cdot 10^{-3}\ \mathrm{sr}^{-1}$ at either site and $|\Delta_{i,j}| > 1.0$ km are excluded.



absolute values may depend on the selection criteria. This is an inherent property of the problem that cannot be avoided: in any intercomparison study the selection of suitable "fair weather"-cases is necessarily somewhat arbitrary – we have tried to mitigate this problem by testing several configurations to find a robust set of criteria. Nevertheless there is certainly not one single set of criteria that can be considered as "absolute truth".

As a by-product we found that in the summer months the mixing layer in Berlin is significantly more extended than in Munich.

In view of our primary question, the representativeness of the spatial distribution of aerosols derived from data of a single ceilometer for a metropolitan area like Berlin or Munich, we conclude that the MLH can be considered as "homogeneous". On the contrary, $\beta_{\mathrm{int}}$ is "less homogeneous" due to the spatiotemporal heterogeneity of aerosol properties and sources. Consequently, the exploitation of denser networks of particle measurements is required, and the deployment of a single ceilometer is not sufficient to characterize the distribution of integrated optical properties as the integrated backscatter. A similar conclusion was e.g. found with respect to clouds over metropolitan areas (Theeuwes et al., 2019). In future, urban scale resolving models (e.g., Maronga et al., 2020) will help to investigate the role of particle transport on the distribution of aerosols in the mixing layer and thus their heterogeneity. With respect to the exploitation of ceilometer data improvements of automated procedures for their calibration are desirable.

## 5 Data availability

Ceilometer data are available on request from the owners.

## 6 Author contributions

MW conducted the study and wrote the paper. AG upgraded the COBOLT-algorithm to make a computational economic retrieval of the MLH feasible, and provided several plots. IM provided data of the DWD-ceilometers including quality control. FM and TR maintained their ceilometers and provided the corresponding data. All authors contributed to the final version of the paper.

*Acknowledgements.* We are grateful to Klaus Riegler (Lufft, OTT HydroMet) for providing ceilometer data. We also thank the responsible persons at DWD that keep their ceilometer-network running. Ceilometers B-TU and B-GR were funded by the Federal Ministry of Education and Research (BMBF), within the framework of Research for Sustainable Development (FONA; www.fona.de), as part of the consortium "Three-dimensional Observation of Atmospheric Processes in Cities" (www.uc2-3do.org), under grant no. 01LP1602.





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
