# Peer review of "On the spatial variability of the regional aerosol distribution as determined from ceilometers"

_Atmospheric Chemistry and Physics, 2020_

## Short Comment (SC1) · 11 May 2020

It is good to see LIDAR data being used for representativity studies, which may help understand better the 4D distribution of aerosol. I wonder if you had an opportunity to look at backscatter at selected heights, and not just integrated backscatter? In our modelling study (https://www.atmos-chem-phys.net/16/6335/2016/), we found that representativity can vary quite a bit between column-integrated properties and profile properties. In particular, representation errors become larger closer to the surface.

I'd also like to point out that, apart from the LIDAR studies you mention, a lot of work on 1) spatio-temporal variability and 2) representation errors has been done using in-situ or satellite data. See the introduction of the aforementioned paper and a follow-up

study from 2017.

---

## Author Comment (AC1) · 13 May 2020

Thank you for reading our manuscript, and your comment and question.

**Your question**

Your question, whether it is possible to intercompare not only the integrated backscatter of the whole mixing layer but also the particle backscatter coefficient of layers (what you call "selected heights"), can be answered with "maybe".

Actually we were thinking about the investigation of elevated layers, e.g. Saharan dust layers, but postponed that for several reasons: (1) the paper is already long, (2) a reliable automatically identification of such layers requires a significant amount of extra work that would have delayed the writing of the paper by an unpredictable (as we know

from "real life") period of time, and (3) we feel that it would be worthwhile to devote a separate paper for this topic (if successful), so that it would be more "visible". In this context an investigation as you suggest would potentially fit. However, the investigation of this topic is likely more time consuming than expected at first glance and thus beyond the scope of our present paper (note, that we have already presented the mean particle backscatter coefficient of the mixing layer in the present paper). It certainly will depend on the vertical resolution of the selected layers, e.g. problems of the interpretation may arise if the layers include the top of the mixing layer, i.e., parts of the mixing layer and of the free troposphere must be averaged. The temporal resolution of the backscatter coefficient profiles (retrieved from the ceilometers) is – depending on the algorithm – up to one hour and maybe must be extended to improve the accuracy of the inversion method; this has to be investigated. In the free troposphere the signal-to-noise ratio of ceilometer signals is typically insufficient for a quantitative intercomparison, or this range is virtually aerosol-free (below the detection limit). Note, that we are using ceilometers, not advanced high power lidars, so a quantitative retrieval above say 5 km is typically impossible. Close to the surface the incomplete overlap prohibits the exploitation of that range. So it can be expected that only a small vertical range can be used for such an intercomparison. We had made the same experience when we tried to validate the water vapor correction in case of Vaisala and Campbell ceilometers, see Wiegner et al. (2019) [https://doi.org/10.5194/amt-12-471-2019]. The remaining vertical range is considerably smaller than the range discussed in your paper "Will a perfect model...", and the interesting range (below 200 m according to your paper) is not included.

So, for the time being it cannot be anticipated whether such an intercomparison results in something of value, and it make no sense to promise something that might turn out to be unrealistic: thus, not a clear "yes" to your question. However, in the framework of our ongoing activities to extend the COBOLT-software we will keep this topic in mind, in particular, as the distance between some of our ceilometers is indeed small, offering unique opportunities.

**Your comment**

We agree with your comment concerning the missing references of studies on the representativeness of scales relevant for e.g. satellite data. Only one has been included in the current version of our manuscript. In the revised version we will include more citations and discuss their relevance for our study.

---

## Referee Comment (RC1) · Anonymous Referee #3 · 22 Jul 2020

The manuscript by Wiegner et al. has the objective to assess the value of using the Mixing Layer Height (MLH) and the Integrated Backscatter (IB) in MLH vertical region, both derived from a network of laser ceilometer operating at a regional scale, to quantify the spatial variability of the aerosol distribution.

The authors present an extensive analysis mainly based on the correlation among spatial differences of the MLH and IB values for two data clusters collected in the two metropolitan areas of Munich and Berlin, using a sufficiently long dataset (2 years approximately depending on the applied data filtering options discussed in the manuscript). For both the clusters, an empirical approach to filter out potential outliers, due to extremes or supposed unphysical values, is also applied and meteorological conditions (i.e. cloudiness and cloud types) are taken into account into the data analy-

sis.

The authors conclude the manuscript asserting that MLH is a more homogenous variable than IB at a regional scale. Discussion on the data homogeneity is based on the spatial variability of both the parameters and on the related uncertainties (also for different retrieval schemes).

The authors' conclusions are a bit vague because they should provide recommendation on the usage of one variable or another with respect to specific applications and this cannot be done without considering the total uncertainty budget for each variable.

I must admit that the manuscript title and the spirit of the manuscript, which seems to investigate in parallel MLH and IB with aim to show which is the best variable to use for assessing the aerosol spatial variability, creates in the reader the expectation to have a final recommendation on which is the best variable to use for investigating the aerosol spatial distribution. My opinion is that different applications needs different input variables. It is undoubted that MLH is a fundamental variable to provide as input to air quality models, for example. Nevertheless, needs of satellite data or forecast models may be different.

I think that the paper could be a bit restructured to become an assessment of what a ceilometer network can or cannot do for the study of the aerosol spatial variability. May be the authors meant to write the paper in this way but an external reader may have a different perception.

Some of the considered aspects are not new for example the uncertainty affecting the retrieval of the IB due to the low signal-to-noise of the ceilometer and to the instability of the calibration constant over the time. These have been already discussed in literature. Similarly, the ceilometer accuracy in capturing the aerosol geometrical properties, such as the MLH, in opposition to the larger uncertainties of retrievals of the aerosol optical properties is already known from the existing literature.

[Figure]

Therefore, my main point for the authors is to adjust the manuscript, in particulate the abstract, the introduction and the summary and conclusions following a different and more concrete style, always referring to the total uncertainty budget and stressing that different variables may be useful for different applications.

A second point is that the authors must also motivate why they use the IB only instead of profiling measurements for their data analysis. To my opinion, the investigation carried out for the IB must be extended to the backscattering profile or to the attenuated backscatter which although not fully quantitative aerosol optical property, is often used for the satellite validation (e.g. CALIPSO data) and also recently used in forecast models and atmospheric reanalysis.

I report below other general and specific comment (line-by-line) afterwards:

I report below other general concerns and specific comment (line-by-line) afterwards:

1. Several times in the manuscript the uncertainty of the MLH and IB values is discussed, and this is a great aspect. Nevertheless, the estimation of the uncertainty for the MLH is mainly related to the algorithmic uncertainty: it seems that in the applied algorithm there other uncertainty contribution which are neglected or not properly considered. For example is the algorithm working on the 15 m raw resolution data? Is an uncertainty of half of the raw resolution, at minimum, used in the MLH estimation? At page 7, lines 8-10, it is written that ".... For this purpose the MLH is smoothed in time ....". Is this smoothing considered in the uncertainty quantification? Which kind of smoothing is applied and it is one of the main reason why the MLH is very homogenous? I think the authors must clarify these aspects.

2. Outliers are considered and treated in the data analysis, very often just using empirical criteria sometimes not motivate by references or specific justifications. Similarly the use of the linear regression method is used without showing any probability density function for the differences of the MLH and IB values. Scatter plots (very small) are used to graphically investigated the results. I think the authors must report the pdfs

instead or along with scatter plots. The pdfs can help to understand where the outliers are and to check the robustness of linear regressions. Otherwise robust linear regression methods may be used.

3. The authors investigated the considered dataset both merging night and daytime data and separating them. My impression is that the investigation of the nocturnal boundary layer is a bit risky considering that fact that the ceilometer is often not sensitive to the static boundary layer height (e.g. when MLH < 220 m) and considering the potential increase in the false positive errors related to an improper gradient attribution in the covered vertical range. The authors are asked to comment on this aspect.

4. The calculation of the ceilometer calibration constant CL reveals an instability over the time, which has been also already pointed out in literature; the authors estimated the calibration instability using a standards deviation (about 14 % considering data from all the stations). The plot shown in Figure 11 clearly shows a quite scattered data distribution: I think the use of standard deviation can underestimate the uncertainty and in general it is not clear how the standard deviation is estimated over the 24 months period. The authors should clarify and improve the robustness of the uncertainty estimation (e.g. showing the pdf and using, if needed, the IQ-range). In addition I think that the use of the same ceilometer type at different measurement sites cannot provided unbiased results although it can reduce the uncertainties related to the presented study: the big difference in the value of CL among the different sites, along with a dependence on seasonal meteorological conditions, is one point against the authors' hypothesis. I ask the authors to reconsider this concept in the different part of the manuscript where it is mentioned.

5. There is a lack of a more general discussion on the reproducibility of the presented approach if applied to other commercial instruments.

Specific comments:

Abstract Line 1: please put "only" at the end of the sentence.

Line 2: "at" instead of "for".

Page 1, line 16: please consider also air quality applications.

Page 1, line 23: put "For example, this.." at the beginning of the sentence.

Page 2, line 3: CALIPSO is an acronym.

Page 2, line 5: rephrase as "... to the topic of the aerosol spatial variability."

Page 2, line 16: "have" instead of "has".

Page 2, line 20: put "in the framework of air quality study" between commas.

Page 3, line 7: ".....we exploit data from (put the number) ceilometers..."

Page 3, line 10-11: rephrase with an appropriate language, please use the concepts of co-location, sampling or representativeness uncertainty.

Page 5: Table 5 could be enriched with all the information discussed at pages 3,5, and 6 to describe the measurement sites, also in order to shorten the text.

Page 7, lines 1-7, the provided description is not fully clear, please improve it.

Page 7, line 12: put "Although" before "such conditions".

Page 8, line 30: put "however" at the beginning of the sentence.

Page 9, line 16-17: the sentence is not clear, please clarify.

Page 9, line 22: some more explanation on the nature or on the reasons for the filtered outliers would be interesting although they are representative of a small fraction of the dataset.

Page 11, line 15: is "mixing Layer Height" the right way to define the nocturnal boundary layer? Please check this carefully.

Page 15, line 7: please show the pdf at the Lindeberg site to demonstrate that the MLH

distribution is narrower than for other sites.

Page 15, line 9: It appears that discussion on the presence of outliers for the M cluster is less extensive than for B cluster.

Page 18, line 1: comparison with literature paper are often reported in the authors' manuscript but it is never mentioned which are the MLH algorithm applied. Please describe them shortly to make more meaningful the comparison with literature results.

Page 22, line 2: "publication in literature".

Page 22, line 8: change the article "a" with "an".

Page 22, line 15: explain better the impact which the incomplete overlap region below 220 m, considered for this study, may have on a comparison with AOD from a use photometer.

Page 23, line 31: check the article ""an".

Page 24, line 12-13: please quantify the number of removed cases.

Page 24, line 25: please check again this sentence.

Page 25, line 3: not sure what's an "aka" constant is.

Page 25, line 3-4: add a reference from existing literature at the end of this sentence.

Page 26, line 1: is the data distribution normal? Please provided more details.

Page 28, liens 17-22: This part seems not to be well integrated into the section. Please check it.

---

## Referee Comment (RC2) · Anonymous Referee #4 · 29 Jul 2020

The paper investigates the spatial variability of the regional aerosol distribution using the so-called ALC (automated low power lidar and ceilometer) signals from two testbeds in Munich and Berlin, Germany. In particular, for each testbed measurements of 24 months from 6 ALCs of the same type (i.e. CHM15k) were analyzed to derive the mixing layer height (MLH) and the integrated backscatter (IB). The intercomparisons of these parameters show that: MLH mean differences are very similar between two sites (i.e. below 50 m) and the correlation coefficient slightly decreases with the distance between the corresponding sites; IB mean differences are of approximately 30 % when individual sites are compared whereas the correlation coefficient shows high and moderate correlation within the Berlin and Munich testbed, respectively.

The measurements coming from the ALCs constitute a remarkable dataset and the

study is of interest. There some points that need to be discussed and/or clarified: -ALC measurement representativeness in characterizing the aerosol spatial variability. The ALC parameters (MLH and IB) are derived using measurements fulfilling specific requirements in terms of the testbed atmospheric conditions (e.g. cloud fraction, rain,  $\beta$ p value, etc..). Thus, the analyzed dataset refers only to a subset of aerosol spatial variability, affecting the sampling representativeness. In other terms: how much the results derived in this work are representative of the aerosol spatial variability in a regional domain? How could you extend these results to the atmospheric conditions not considered in this work? This aspect needs to be exhaustively discussed in the paper. - ALC instrumental terms affecting the ALC parameters. It could be of interest to evaluate the contribution of the ALC instrumental variability to the measured spatial variability. In particular, the effect of the overlap function on IB should be more detailed (e.g. how the assumption of a constant  $\beta$ p in the region of incomplete overlap can affect the variability? Is this assumption valid for different environments, urban and rural, and different season?). The effect of the calibration constant (CL) temporal variability is taken into account in the paper. However, more details should be provided regarding the CL uncertainty and its effect on IB variability.

The impression is that only partial information about the aerosol variability has been exploited from the potential information provided by the large number of ALCs used. The results obtained in terms of horizontal variability of MLH are similar to other works and the conclusion that MLH parameter is more homogenous than IB at a regional scale seems self-evident. In summary, the authors should well define the scientific context and the objectives of the work to better characterize the obtained results, highlighting their novelty and relevancy. Thus, I recommend the publication of the manuscript after major revisions, according to the following observations.

Major comments:

Introduction
- The introduction should give an overview of the studies regarding the aerosol spatial variability using ground-based and satellite-borne passive sensor measurements as well as airborne and satellite-borne lidar measurements. Which are the limits and the benefits of using ALC datasets (e.g. temporal and spatial scales)? Please add some discussions and references.

- Page 2, Lines 12-13: recent works used ALC-derived backscatter also to derive aerosol optical depth and aerosol volume and mass (e.g. Dionisi et al., 2018 and Diemoz et al., 2019). Please add some references.

**Section 2**

- In this section, a characterization of the two testbeds (e.g. predominant local atmospheric circulation, topography, etc...) should be provided. This could be of help to interpret the obtained result and to establish the limit of using ALC measurements for aerosol variability studies. Figure 9 could be inserted in this section.

- Page 3, Lines 24-26: the work of Hervo et al., 2016 shows that the manufacturer overlap function cannot account for changes over time and that the derived correction is temperature-dependent. This could significantly affect the ALC profile in the first hundreds of meters. Did the authors take into account this problem?

- Page 6, Line 11: Is this only because B-PO is closer to Berlin in respect of B-LI? Please provide some references or a more detailed explanation.

- Page 6, Lines 14-17: I don't understand if the periods mentioned above are measurement gaps (if this is the case, these are not a few hours) or something else. Please explain or rephrase.

- Page 6, Line 18: the temporal resolution of MLH values (i.e. 2 min, 720 values per day) is defined on page 22 line 23 but it should be clearly defined in this section.

- Page 7, Lines 14-17: please add some details about this overlap correction. Is this correction applied also to compute the aerosol IB?

**ACPD**
**Section 3**

This section should be re-organized to better highlight the results of the study. In particular, the authors could point out if this ALC set-up and the conducted analyses allow characterizing different environmental conditions (e.g. urban vs urban, urban vs rural, rural vs rural, day vs night) or not. This implies the description of the considered sample, the characterization of the impact of the considered selecting criteria, the identification and evaluation of all the potential factors contributing to the resulting variability. This information is scattered throughout the sub-sections but it is not easy to put it together.

**Section 3.1**

It is not clear to me if the cloud cover parameter is used with the ALC raw temporal resolution. Please clarify this point. Moreover, to better characterize the use and the representativeness of ALC measurements in this context (i.e. aerosol spatial variability), it could be of interest to add in this section a resuming table that shows the number of the available cases associated to the different filtering criteria considered to retrieve MLH and IB. Please also clarify why the reference configuration has been set as the 'reference.

**Section 3.2**

- Fig. 3 and 4. The relative frequency distribution plots provide potential information regarding urban vs rural conditions (asymmetric shape). This could be stressed in the text. Do the authors think that further analyses of the distribution shape could help in the characterization of the different sites in terms of variability? The scatter plots below the diagonal seem to be characterized by a large number of low MLH values concerning high MLH values. Is the linear regression weighted for MLH?

- Fig. 5, 6, and section 3.2.3: the correlation decreases with the distance up to 40-50 km then it seems to be constant. This could mean that at a certain distance from the
city, rural conditions are more homogeneous and less affected by urban conditions. Please clarify.

Section 3.3

- Fig 7 and 8. Please specify the bi-monthly number of points used to compute the diurnal cycle for each testbed.

- Fig. 9: I'm not sure that this figure is appropriate in this section.

Section 3.4

- Page 22, Eq.1: as already mentioned (see general comment), the assumption of a constant  $\beta p$  in the region of incomplete overlap, in particular in urban environments, could lead to biased values of IB. In the work of Barnaba et al. 2010, the lidar profiles in the incomplete overlap region were obtained by a linear fit of the first two valid lidar points. A sensitivity study considering also this approach could evaluate how the different assumptions affect IB values.

- Page 22, line 20: please explain the choice of LR=45 sr.

- Page 24, lines 25-29: strengthening the criteria (reducing the sample population) increase the correlation coefficient as more homogenous conditions are considered. The information on the impacts of the different selected criteria could help in characterizing the sample and the limit of the analysis. Please highlight this point in the text.

- Figure 11: please explain the large amount of high values of CL between 250 and 350 days.

- Page 26, lines 1-4: please explain why the authors decided to use the averaged CL over 24 months instead of the daily CL. How much this choice affects the IB variability?

- Page 26, lines 19-27: see the comment of Page 24, lines 25-29.

- Page 29, lines 10-11: these factors are crucial for the aerosol spatial variability. They
should be mentioned not only at the end of the paper but also at the beginning, to contextualize the objectives and limits of the considered approach.

Section 4

- Page 29, lines 19-20: the same type of instrument does not prevent the fact that each ALC has a different variability of instrumental parameters such as the CL term and the overlap function. This aspect should be pointed out in the text.

- Page 31, lines 0-3: the selection of arbitrary criteria to avoid 'fair-weather cases' deals with the problem of defining the sampling representativeness. This should be discussed in this section.

- Page 31, lines 8-11: the result of this study could suggest that a denser ALC network could be required in metropolitan areas, whereas a smaller number of ALCs could be representative of rural areas.